# Last-Iterate Convergence for Generalized Frank-Wolfe in Monotone Variational Inequalities

**Zaiwei Chen**
Purdue IE
West Lafayette, IN 47907
chen5252@purdue.edu

**Eric Mazumdar**
Caltech CMS
Pasadena, CA 91125
mazumdar@caltech.edu

## Abstract

We study the convergence behavior of a generalized Frank-Wolfe algorithm in constrained (stochastic) monotone variational inequality (MVI) problems. In recent years, there have been numerous efforts to design algorithms for solving constrained MVI problems due to their connections with optimization, machine learning, and equilibrium computation in games. Most work in this domain has focused on extensions of simultaneous gradient play, with particular emphasis on understanding the convergence properties of extragradient and optimistic gradient methods. In contrast, we examine the performance of an algorithm from another well-known class of optimization algorithms: Frank-Wolfe. We show that a generalized variant of this algorithm achieves a fast $\mathcal{O}(T^{-1/2})$ last-iterate convergence rate in constrained MVI problems. By drawing connections between our generalized Frank-Wolfe algorithm and the well-known smoothed fictitious play (FP) from game theory, we also derive a finite-sample convergence rate for smoothed FP in zero-sum matrix games. Furthermore, we demonstrate that a stochastic variant of the generalized Frank-Wolfe algorithm for MVI problems also converges in a last-iterate sense, albeit at a slower $\mathcal{O}(T^{-1/6})$ convergence rate.

## 1 Introduction

A constrained monotone variational inequality (MVI) problem consists of solving for an $x^* \in \mathcal{X} \subseteq \mathbb{R}^d$ such that

$$\max_{s \in \mathcal{X}} (x^* - s)^\top F(x^*) \leq 0,$$

where $F : \mathcal{X} \to \mathbb{R}^d$ is a monotone operator [56, 37] and $\mathcal{X}$ is a convex set. MVIs arise in many foundational and emerging problems. In particular, many problems in optimization [10, 1, 39], equilibrium computation [43, 42], reinforcement learning [57, 38], and learning in games [13] can be formulated as MVI problems.

Due to their wide applicability, recent years have seen significant advances in developing efficient algorithms to solve these problems. Despite the structure provided by the monotone mapping, MVI problems are well-known to be challenging to solve, as simple first-order algorithms may diverge or exhibit complex limiting behaviors such as chaos [3, 27, 18]. This has motivated the analysis of algorithms such as the extragradient method [37], the optimistic gradient method [51], and the Halpern iteration method [19]. These algorithms, particularly the extragradient and optimistic gradient methods—which can be viewed as approximating proximal point algorithms [46]—have been the focus of numerous recent works, with matching upper and lower bounds established under various assumptions about the feasible set $\mathcal{X}$ and the operator $F(\cdot)$ [29, 30, 15, 26, 41]. Due to their connection with gradient descent and various extensions in convex optimization, these algorithms

have garnered the most attention in the literature, with recent breakthroughs in the constrained regime where $\mathcal{X}$ is a compact convex set [15]. See Section 1.1 for more details about related work.

In this paper, we take an orthogonal approach by analyzing the performance of another optimization algorithm for solving constrained MVI problems: Frank-Wolfe (FW) [21]. Although the FW algorithm was first proposed for solving MVI problems decades ago [32] and has been analyzed in the context of min-max optimization problems [24], its convergence rate for general MVI problems remains less understood. To address this, we provide an analysis of a smoothed version of FW for solving constrained MVI problems, even considering the case where one only has access to noisy estimates of the operator $F(\cdot)$. This case is particularly relevant to problems in machine learning [23], distributionally robust optimization [5, 61], and learning in games [25, 43].

## 1.1 Related Literature

There is a rich literature analyzing MVI problems and their solutions [8, 54]. In this section, we give an overview of the most related works.

**Gradient-Based Methods.** Work on solving MVI problems has largely focused on understanding the behavior of gradient-based algorithms due to their connection with gradient descent in optimization, though other algorithms have also been proposed in the literature (see, e.g., [60]). Although straightforward generalizations of gradient descent can fail in MVI problems [42, 43], proximal point algorithms [52] and related methods such as the extragradient [37] and optimistic gradient [18] algorithms have been shown to provide much stronger convergence guarantees. Specifically, [26] showed that the extragradient algorithm achieves a tight $\mathcal{O}(T^{-1/2})$ last-iterate convergence rate for the smooth convex-concave saddle-point problem (a special case of MVI). In [17], the authors also studied saddle-point problems and proposed an algorithm called mirror-prox conditional gradient sliding, which comes with strong complexity guarantees. However, their analysis required a strong concavity assumption on the objective function, which corresponds to a strong monotonicity assumption in the variational inequality formulation. In [29], the authors studied unconstrained variational inequality problems and showed that the extragradient algorithm achieves an $\mathcal{O}(T^{-1})$ last-iterate convergence rate under the assumptions of monotonicity and Lipschitz continuity of $F(\cdot)$. Finally, in [15], the authors established the tight $\mathcal{O}(T^{-1/2})$ last-iterate convergence of both the extragradient and optimistic gradient descent-ascent algorithms for constrained MVIs. However, the approach in [15] relies on computer-aided proofs, whereas our proof uses a natural Lyapunov argument.

**The Halpern Iteration Method.** The Halpern iteration method was originally proposed to find the fixed points of non-expansive mappings [31]. More recently, algorithms based on the Halpern iteration have been applied to solving MVI problems with a fast $\mathcal{O}(T^{-1})$ convergence rate [19, 58]. However, to the best of our knowledge, there are no results showing that the Halpern iteration method, or the algorithms proposed in [19, 58], have provable convergence in the stochastic setting.

**Frank-Wolfe Methods.** The closest work to ours in this area is [24], which analyzes FW in deterministic convex-concave saddle-point problems. They prove last-iterate convergence rates by making curvature assumptions on the operator $F(\cdot)$ and on the underlying space $\mathcal{X}$ (i.e., assuming $\mathcal{X}$ is strongly convex or $F(\cdot)$ is strongly monotone). Additionally, they show a slow convergence rate for FW on polytopic sets without these curvature assumptions. In contrast, we analyze a smoothed version of FW, also known as the generalized conditional gradient algorithm [2, 12], in MVIs. We demonstrate that this generalized version of FW achieves fast convergence without imposing strong curvature assumptions on either the operator or the underlying space. Crucially, the smoothing technique allows us to bypass issues that arise with vanilla FW in saddle-point problems. Another application of FW in MVIs is presented in [35], where FW is used to compute the iterates in mirror-prox for MVIs.

**Stochastic Monotone Variational Inequality Problems.** There has been considerable recent work on solving stochastic MVI problems, where one can only obtain noisy estimates of the operator $F(\cdot)$. Such problems arise in multi-agent reinforcement learning [62] and distributionally robust supervised learning [61], among other domains. However, the literature is sparser for this class of problems, particularly regarding last-iterate convergence in constrained problems. Under curvature assumptions (on $F(\cdot)$ and/or $\mathcal{X}$), stronger guarantees (both in expectation and with high probability) exist for variants of extragradient and optimistic gradient algorithms [8, 28, 44]. Inspired by recent results using FW for stochastic optimization [45, 20], we extend our smoothed FW algorithm to

stochastic MVI problems and, to the best of our knowledge, provide the first last-iterate convergence guarantee for an algorithm in constrained stochastic MVI problems without curvature assumptions on the monotone operator. While the $\mathcal{O}(T^{-1/6})$ rate of convergence we derive for this algorithm is slower than the known $\mathcal{O}(T^{-1/2})$ convergence rate for the averaged iterates of the mirror-prox algorithm [36], it is important to note that the mirror-prox algorithm has been shown to diverge in stochastic monotone problems [16]. Furthermore, in MVI problems, it has been demonstrated that the averaged iterates can exhibit fundamentally faster convergence rates than the last iterate [26].

## 1.2 Our Contributions

We introduce and analyze a generalized FW algorithm for solving MVI problems. This algorithm is a natural extension of the classic smoothed fictitious play (FP) algorithm for learning in games [22], applied to monotone games and, by extension, to MVI problems. We show that the algorithm achieves a fast last-iterate convergence rate of $\mathcal{O}(T^{-1/2})$, matching the rates of optimistic gradient and extragradient algorithms. As a consequence of our analysis, we derive a finite-time bound for smooth FP in finite zero-sum games.

We also consider the case of stochastic MVI problems, where only noisy estimates of $F(\cdot)$ are available. We demonstrate that, by designing estimators similar to those used in stochastic FW for optimization, it is possible to achieve last-iterate convergence in constrained MVI problems using generalized FW, though at a slower rate of $\mathcal{O}(T^{-1/6})$. Although this rate is not optimal, it appears to be the first last-iterate convergence rate for solving constrained stochastic MVI problems without assuming strong curvature properties of the operator $F(\cdot)$ or the set $\mathcal{X}$. Indeed, previous algorithms have provided last-iterate guarantees either in the unconstrained setting [14] or under curvature assumptions on $F(\cdot)$, such as strong (quasi)-monotonicity or coercivity [8].

## 2 Problem Formulation

Let $F : \mathcal{X} \to \mathbb{R}^d$ be a (possibly nonlinear) operator, where $\mathcal{X}$ is a convex and compact subset of $\mathbb{R}^d$. The associated variational inequality problem consists of solving for an $x^* \in \mathcal{X}$ such that

$$\max_{s \in \mathcal{X}} (x^* - s)^\top F(x^*) \leq 0. \tag{1}$$

Although such problems canonically arise in optimization [54] and machine learning [39], where the operator $F(\cdot)$ is usually the gradient of some objective function, the formulation is general enough to capture other problems such as reinforcement learning [38] and learning in games [13]. Next, we provide two illustrative examples.

**Example 1: The Policy Evaluation Problem in Reinforcement Learning.** Consider an infinite horizon discounted Markov decision process (MDP) with a finite state space $\mathcal{S}$, a finite action space $\mathcal{A}$, a set of action-dependent transition probability matrices $\{P_a \in \mathbb{R}^{|\mathcal{S}| \times |\mathcal{S}|} \mid a \in \mathcal{A}\}$, a reward function $\mathcal{R} : \mathcal{S} \times \mathcal{A} \to \mathbb{R}$, and a discount factor $\gamma \in (0, 1)$. The transition probabilities and the reward function are unknown to the agent. Given a policy $\pi : \mathcal{S} \to \Delta(\mathcal{A})$, where $\Delta(\mathcal{A})$ denotes the probability simplex on $\mathcal{A}$, its value function $V^\pi \in \mathbb{R}^{|\mathcal{S}|}$ is defined as

$$V^\pi(s) = \mathbb{E}_\pi \left[ \sum_{t=0}^\infty \gamma^t \mathcal{R}(S_t, A_t) \mid S_0 = s \right]$$

for all $s \in \mathcal{S}$, where $\mathbb{E}_\pi[\cdot]$ means that the actions are chosen according to the policy $\pi$. The policy evaluation problem in reinforcement learning refers to the problem of estimating $V^\pi$ for a given policy $\pi$ [57]. To solve this problem, it has been shown that $V^\pi$ is the unique solution of a fixed-point equation known as the Bellman equation $V = \mathcal{T}^\pi(V)$, where $\mathcal{T}^\pi : \mathbb{R}^{|\mathcal{S}|} \to \mathbb{R}^{|\mathcal{S}|}$ is the Bellman operator [50]. Therefore, solving the policy evaluation problem is equivalent to solving the unconstrained variational inequality problem $V - \mathcal{T}^\pi(V) = 0$.

**Example 2: Multi-Player Convex-Concave Games.** Consider an $n$-player game where each player $i \in \{1, 2, \cdots, n\}$ has a compact convex action set $\mathcal{X}_i \subseteq \mathbb{R}^{d_i}$ and a loss function $f_i : \prod_{j=1}^n \mathcal{X}_j \to \mathbb{R}$ such that $f_i(x_i, x_{-i})$ is convex in $x_i$ for all $x_{-i} \in \prod_{j \neq i} \mathcal{X}_j$. Such games have been well analyzed

in the literature in economics [53] and more recently in machine learning [42]. Solving for a Nash equilibrium [47] in this game can be formulated as solving for a point $x^* \in \mathcal{X} := \prod_{j=1}^{n} \mathcal{X}_j$ such that $\max_{s \in \mathcal{X}} (x^* - s)^\top F(x^*) \leq 0$, where

$$F(x) = [\nabla_{x_1} f_1(x_1, x_{-1}), \cdots, \nabla_{x_n} f_n(x_n, x_{-n})], \quad \forall\, x \in \mathcal{X}.$$

Indeed, by convexity, we have for any $x \in \mathbb{R}^d$ that

$$\max_{s \in \mathcal{X}} (x - s)^\top F(x) \geq \sum_{i=1}^{n} \{f_i(x_i, x_{-i}) - \min_{s_i \in \mathcal{X}_i} f_i(s_i, x_{-i})\} \geq 0,$$

where the equality is achieved if and only if a joint strategy $x^* = (x_1^*, \cdots, x_n^*)$ satisfies $f_i(x_i^*, x_{-i}^*) = \min_{s_i \in \mathcal{X}_i} f_i(s_i, x_{-i}^*)$ for all $i$, which implies that $x^*$ is a Nash equilibrium of the game.

An important class of variational inequality problems is MVI problems where the operator $F(\cdot)$ is monotone over $\mathcal{X}$. Note that an operator $F : \mathcal{X} \to \mathbb{R}^d$ is said to be monotone if and only if

$$(F(x_1) - F(x_2))^\top (x_1 - x_2) \geq 0, \quad \forall\, x_1, x_2 \in \mathcal{X}. \tag{2}$$

Despite this additional structure, designing algorithms for solving constrained MVI problems efficiently and with strong convergence guarantees has been an open problem until recently [15], with most work focused on analyzing approximations of proximal point algorithms such as extragradient and optimistic gradient approaches [29, 30, 15].

Further generalizations of the MVI problem that are of particular interest for applications in machine learning are stochastic MVI problems, where one only has access to a noisy estimator of $F(x)$. Such situations arise in, e.g., reinforcement learning (where the agent learns by interacting with the environment) [57] and problems of distributionally robust optimization where one seeks to solve a zero-sum game using mini-batches to estimate gradients [61, 48, 16].

The rest of this paper is organized as follows. To motivate the generalized FW algorithm for MVI problems, we first present the smoothed FP, a canonical algorithm for learning in games, which we view as the instantiation of generalized FW in zero-sum matrix games. We then introduce the generalized FW algorithm for solving MVI problems and present its last-iterate convergence rate. Moving to the stochastic setting, we propose a stochastic variant of the generalized FW algorithm also with last-iterate convergence guarantees. Notably, the algorithm employs a two-timescale structure, where we construct a variance-reduced estimator of $F(x)$ on the fast timescale and implement the generalized FW algorithm on the slow timescale.

## 3 Warm-Up: Smoothed Fictitious Play

In this section, we present the problem of finding a Nash equilibrium of a zero-sum game and reformulate it as an MVI problem. In addition, we present the smoothed FP algorithm for zero-sum games, which also motivates our algorithm for the MVI problem (1) in the next section.

Consider a two-player finite zero-sum game where the set of pure strategies[1] for player $i$ (where $i \in \{1, 2\}$) is denoted by $\mathcal{A}^i$. When players play over mixed strategies, we can write this game as the min-max optimization problem $\min_{\pi^2 \in \Delta(\mathcal{A}^2)} \max_{\pi^1 \in \Delta(\mathcal{A}^1)} (\pi^1)^\top R \pi^2$, where $R \in \mathbb{R}^{|\mathcal{A}^1| \times |\mathcal{A}^2|}$ is the payoff matrix. A canonical measure of the performance of algorithms for learning in such games is the Nash gap, which measures how far each player is from their best response:

$$\text{NG}(\pi^1, \pi^2) = \max_{\bar{\pi}^1 \in \Delta(\mathcal{A}^1)} (\bar{\pi}^1)^\top R \pi^2 - \min_{\bar{\pi}^2 \in \Delta(\mathcal{A}^2)} (\pi^1)^\top R \bar{\pi}^2.$$

Suppose that a pair of strategies $(\pi_*^1, \pi_*^2)$ satisfies $\text{NG}(\pi_*^1, \pi_*^2) = 0$. Then, each player is playing the best response to their opponent's strategy, thereby having no incentive to deviate from their current strategy. This situation defines a Nash equilibrium [47].

Solving for a Nash equilibrium in such games has been a focus of interest in economics and the literature on learning in games, dating back to [59]. One of the most canonical algorithms for learning in games from that literature is FP, where players play the best responses to the empirical history

---

[1] We will use strategy and policy interchangeably.

of their opponents' actions. Subsequently, a generalization of that algorithm, smoothed FP, was introduced as it was found to be a better model of human play, accounting for "trembling-hand" strategies in games [22]. In smoothed FP, players again keep track the empirical history of their opponents' play but instead sample an action from a smoothed best-response strategy rather than playing the exact best response.

More concretely, for any $x \in \mathbb{R}^d$ such that $x_i \geq 0$ for all $i$, let $\nu(x) = -\sum_{i=1}^{d} x_i \log(x_i)$ be the entropy function [55]. Given $i \in \{1, 2\}$ and $a^i \in \mathcal{A}^i$, we use $e(a^i)$ to denote the $|\mathcal{A}^i|$-dimensional vector with its $a^i$-th entry being one and zero everywhere else. Then, players make use of the algorithm presented in Algorithm 1 for repeatedly playing the finite zero-sum games.

---

**Algorithm 1** Smoothed Fictitious Play (from Player 1's perspective)

---

1: **Input:** Integer $T$, temperature $\tau > 0$, and initialization $\pi_0^2 \in \Delta(\mathcal{A}^2)$
2: **for** $t = 0, 1, \cdots, T - 1$ **do**
3:     $v_t^1 = \arg\max_{v \in \Delta(\mathcal{A}^1)} \{v^\top R \pi_t^2 + \tau \nu(v)\}$
4:     Play $A_t^1 \sim v_t^1(\cdot)$ and observe $A_t^2$
5:     $\pi_{t+1}^2 = \pi_t^2 + \frac{1}{t+1}(e(A_t^2) - \pi_t^2)$
6: **end for**

---

In smoothed FP, with an arbitrary initial estimate of the opponent's policy $\pi_0^2$, in each round, player 1 plays the smoothed best response to its latest estimate of the opponent's policy (cf. Algorithm 1 Line 4), and updates the estimate $\pi_t^2$ according to Algorithm 1 Line 5, which is an iterative way of computing the empirical average of the opponent's historical strategies.

Despite its canonical nature and its connection to classic algorithms in online learning, such as Follow-The-Regularized-Leader (FTRL) [40], the algorithm lacks a finite-time convergence rate guarantee, although it has been well analyzed in its continuous-time limit [33, 6]. To connect with FTRL, suppose that player 1 observes $v_t^2$ and does not know $R$, but has payoff-based feedback of the form $r_t$ such that $\mathbb{E}[r_t \mid A_t^1, A_t^2] = R(A_t^1, A_t^2)$. In this case, smoothed FP reduces to FTRL or forms of FTRL with bandit feedback due to the linear structure of the losses. However, smoothed FP assumes an unusual feedback structure (for online learning algorithms) in which each player is assumed to know the payoff matrix $R$, but can only observe the realized actions of their opponent, not the entire strategy $v_t^2$. Therefore, previous approaches for analyzing FTRL do not apply, and, to the best of our knowledge, finite-time analysis of smoothed FP is still lacking in the literature, although it has been shown to be asymptotically no-regret [6].

The following convergence rate of Algorithm 1 follows as a consequence of our more general results of generalized FW. Since we are dealing with a finite game, we assume, without loss of generality, that $\max_{a^1 \in \mathcal{A}^1, a^2 \in \mathcal{A}^2} |R(a^1, a^2)| \leq 1$.

**Theorem 3.1.** *Suppose that both players use smoothed FP in finite zero-sum games and $\tau \in (0, 1]$. Then, we have for any $t \geq 0$ that*

$$\mathbb{E}[NG(\pi_t^1, \pi_t^2)] \leq \frac{4\sqrt{|\mathcal{A}^1| + |\mathcal{A}^2|}}{t + 1} + \frac{36|\mathcal{A}^1||\mathcal{A}^2|\log(t+1)}{\tau(t+1)} + \tau \log(|\mathcal{A}^1||\mathcal{A}^2|).$$

The proof of Theorem 3.1 is presented in Appendix A. In view of Theorem 3.1, given a time horizon $T$, by choosing $\tau = \mathcal{O}\left(T^{-1/2}\right)$, we have an $\tilde{\mathcal{O}}\left(T^{-1/2}\right)$ rate of convergence of the empirical history of the play. Equivalently, we have the following iteration complexity for the algorithm.

**Corollary 3.1.1.** *To achieve $\mathbb{E}[NG(\pi_t^1, \pi_t^2)] \leq \epsilon$, the iteration complexity is $\tilde{\mathcal{O}}(|\mathcal{A}^1||\mathcal{A}^2|/\epsilon^2)$.*

To identify Algorithm 1 for zero-sum games as a special case of generalized FW for MVI problems, observe that the Nash gap $NG(\cdot, \cdot)$ can be rewritten as

$$NG(x_1, x_2) = \max_{s \in \mathcal{X}} (x - s)^\top F(x), \tag{3}$$

where $x = (x_1, x_2) \in \mathcal{X} := \Delta(\mathcal{A}^1) \times \Delta(\mathcal{A}^2)$ and $F(x) = Mx$ with the matrix $M$ defined as $M = [0^{|\mathcal{A}^1| \times |\mathcal{A}^2|}, -R; R^\top, 0^{|\mathcal{A}^2| \times |\mathcal{A}^1|}]$. Since $M + M^\top = 0$, it is clear that $F(\cdot)$ is a monotone

operator. In addition, when both players follow smooth FP as presented in Algorithm 1, the joint update equation can be equivalently written as

$$s_t = \arg\min_{s \in \mathcal{X}} \{s^\top F(x_t) + \tau f(s)\}, \tag{4}$$

$$x_{t+1} = x_t - \alpha_t(x_t - s_t + w_t), \tag{5}$$

where $f(s) = -\nu(s_1) - \nu(s_2)$ for any $s = (s_1, s_2) \in \mathcal{X}$, and $w_t$ is a zero-mean random variable. In smoothed FP, the random variable $w_t$ corresponds to the difference between the softmax distribution (cf. Algorithm 1 Line 3) and a sample from the softmax distribution (cf. Algorithm 1 Line 4). In view of Eqs. (3), (4), and (5), we see that smoothed FP for zero-sum matrix games is simply a generalization of FW for MVI problems. Although this algorithm has been well analyzed in the optimization literature (i.e., when $F(\cdot)$ is the gradient of some objective function) [2, 11, 12], it has yet to be analyzed, to the best of our knowledge, in the context of more general MVI problems. In the next section, we show that this algorithm has strong convergence properties.

## 4 Generalized Frank-Wolfe for Monotone Variational Inequalities

Motivated by the smoothed FP for zero-sum games, we next present our algorithm and convergence guarantees for solving general MVI problems.

---

**Algorithm 2** Generalized Frank-Wolfe for Monotone Variational Inequalities

---

1: **Input:** Integer $T$, tunable parameter $\tau > 0$, and initialization $x_0 \in \mathbb{R}^d$
2: **for** $t = 0, 1, \cdots, T - 1$ **do**
3:     $s_t = \arg\min_{s \in \mathcal{X}} \{s^\top F(x_t) + \tau f(s)\}$
4:     $x_{t+1} = x_t - \alpha_t(x_t - s_t + w_t)$
5: **end for**

---

In Algorithm 2 Line 3, the function $f : \mathcal{X} \to [0, \infty)$ serves as a regularizer (analogous to the entropy function in smoothed FP), for which we impose the following requirement.

**Condition 4.1.** The function $f(\cdot)$ is continuously differentiable and $\sigma_f$-strongly convex for some $\sigma_f > 0$. In addition, $\lim_{x \to \partial \mathcal{X}} \|\nabla f(x)\|_2 = +\infty$, where $\partial \mathcal{X} = \mathcal{X} \setminus \mathrm{relint}_{\mathbb{R}^d} \mathcal{X}$ denotes the boundary of the convex compact subset $\mathcal{X}$ of $\mathbb{R}^d$.

Differentiability and strong convexity are standard requirements when choosing regularizers. The condition that $\lim_{x \to \partial \mathcal{X}} \|\nabla f(x)\|_2 = +\infty$ ensures that the generalized FW direction $s_t$ from Algorithm 2 Line 3 always lies in the relative interior of $\mathcal{X}$. These conditions are satisfied by, e.g., the sum of negative entropies when the compact convex set $\mathcal{X}$ is the product of probability simplices. Note that when $\tau = 0$, the algorithm recovers the vanilla version of FW analyzed in [24] for saddle point problems. Although the use of regularization precludes the use linear minimization oracles (LMOs), which is one of the main features that make FW algorithms so appealing [34, 49], we remark that it does not add additional complexity to the algorithm when compared to projected extragradient and optimistic gradient methods. Specifically, note that the subproblem that appears in Algorithm 2 Line 3 is a strongly convex optimization problem and can be solved efficiently or even admits closed-form solutions. For example, when $\mathcal{X}$ the probability simplex and $f(\cdot)$ is the negative entropy, the FW direction $s_t$ is the softmax operator.

To derive our convergence guarantees, we impose the following assumptions on the operator $F(\cdot)$ and the stochastic process $\{w_t\}$.

**Assumption 4.1.** The operator $F(\cdot)$ is Lipschitz continuous, i.e., there exists $L_F > 0$ such that

$$\|F(x_1) - F(x_2)\|_2 \leq L_F \|x_1 - x_2\|_2, \quad \forall x_1, x_2 \in \mathcal{X}.$$

**Assumption 4.2.** The operator $F(\cdot)$ has a Lipschitz continuous Jacobian matrix $J(\cdot)$, i.e., there exists $L_J > 0$ such that

$$\|J(x_1) - J(x_2)\|_2 \leq L_J \|x_1 - x_2\|_2, \quad \forall x_1, x_2 \in \mathcal{X}.$$

In zero-sum games, due to the linear structure of $F(\cdot)$, the Jacobian matrix is the zero matrix. Therefore, Both Assumptions 4.1 and 4.2 are automatically satisfied. In optimization, $F(\cdot)$ is the

gradient of the objective function that we aim to optimize, and Assumptions 4.1 and 4.2 are equivalent to assuming the smoothness of the objective function [4] and the Lipschitz continuity of the Hessian matrix.

**Assumption 4.3.** It holds for all $t \geq 0$ that (1) $s_t - w_t \in \mathcal{X}$, (2) $\mathbb{E}[w_t \mid \mathcal{F}_t] = 0$, (3) $\mathbb{E}[\|w_t\|_2^2 \mid \mathcal{F}_t] \leq \sigma_w$, where $\sigma_w > 0$ and $\mathcal{F}_t$ is the $\sigma$-algebra generated by $\{x_0, w_0, w_1, \cdots, w_{t-1}\}$.

When $\sigma_w = 0$, Algorithm 2 is a deterministic algorithm. More generally, we allow for this additive martingale difference noise to capture the potential stochasticity in choosing the FW direction, which is present in, e.g., smoothed FP, due to sampling an action according to the smoothed best response.

To state our main result for generalized FW in MVI problems, the following notation is needed. Let $D_\mathcal{X} = \max_{x \in \mathcal{X}} \|x\|_2$, $\bar{F} = \max_{x \in \mathcal{X}} \|F(x)\|_2$, and $\bar{f} = \max_{x \in \mathcal{X}} f(x)$, all of which are well defined and finite due to the Weierstrass extreme value theorem because $F(\cdot)$ and $f(\cdot)$ are continuous functions and $\mathcal{X}$ is a compact set. Next, we present the convergence guarantee on the iterates of Algorithm 2 when using stepsizes of various decay rates.

**Theorem 4.1.** *Consider $\{x_t\}$ updated according to Algorithm 2. Suppose that $F(\cdot)$ is a monotone operator on $\mathcal{X}$, and Assumptions 4.1, 4.2, and 4.3 are satisfied. Then, when the regularizer $f(\cdot)$ satisfies Condition 4.1, we have for all $t \geq 0$ that*

$$\mathbb{E}\left[\max_{s \in \mathcal{X}} (x_t - s)^\top F(x_t)\right] \leq \begin{cases} 2D_\mathcal{X}\bar{F}(1-\alpha)^t + c_1\alpha + \tau\bar{f}, & \text{when } \alpha_t \equiv \alpha \leq 1, \\ \dfrac{2D_\mathcal{X}\bar{F}}{t+1} + \dfrac{c_1\log(t+1)}{t+1} + \tau\bar{f}, & \text{when } \alpha_t = \dfrac{1}{t+1}, \end{cases}$$

*where $c_1 = (L_F + L_F^2/(2\tau\sigma_f) + D_\mathcal{X}L_J)(\sigma_w + 4D_\mathcal{X}^2)$.*

The proof of Theorem 4.1 is presented in Appendix B.2. Note that for a given time horizon $T$, choosing $\tau = \mathcal{O}(T^{-1/2})$ results in an overall $\tilde{\mathcal{O}}(T^{-1/2})$ last-iterate convergence rate to a solution to the MVI problem. The problem-dependent constants $D_\mathcal{X}$, $\bar{F}$, and $\bar{f}$ appear additively or multiplicatively in the bound but do not impact the overall $\tilde{\mathcal{O}}(T^{-1/2})$ rate of convergence.

**Corollary 4.1.1.** *To achieve $\mathbb{E}[\max_{s \in \mathcal{X}} (x_t - s)^\top F(x_t)] \leq \epsilon$, the iteration complexity is $\tilde{\mathcal{O}}(\epsilon^{-2})$.*

This convergence rate matches the last-iterate convergence rate recently proved for extragradient and optimistic gradient algorithms for constrained MVI problems [15]. In contrast to the analyses of those algorithms which requires computer-aided proofs such as the sum of squares programming [15] or performance estimation problems [29], our proof follows from a simple Lyapunov argument on the regularized gap $V(\cdot)$, which is defined as

$$V(x) = \max_{s \in \mathcal{X}} \left\{ (x - s)^\top F(x) - \tau f(s) \right\}. \tag{6}$$

A key step in proving Theorem 4.1 is the following lemma, which shows the smooth evolution of the generalized FW directions $\{s_t\}$. For notation convenience, let $s(x) = \arg\min_{s \in \mathcal{X}} \{s^\top F(x) + \tau f(s)\}$ for all $x \in \mathcal{X}$.

**Lemma 4.1.** *It holds for all $x_1, x_2 \in \mathcal{X}$ that $\|s(x_1) - s(x_2)\|_2 \leq \frac{L_F}{\tau\sigma_f}\|x_1 - x_2\|_2$.*

The proof of Lemma 4.1 is presented in Appendix B.1. As a last comment, note that while the $\tilde{\mathcal{O}}(T^{-1/2})$ convergence rate is known to be tight for extragradient and optimistic gradient algorithms [15, 25] (since they both can be seen as instantiations of $p$-stationary canonical linear iterative algorithms), it is unclear whether this rate is tight for FW-type algorithms. We leave further explorations of fundamental lower bounds to future work. We also carried out experiments to numerically compare the performance of our algorithm with those proposed in the literature. Due to space limitation, the results are reported in Appendix D.

## 5 Generalized Frank-Wolfe for Stochastic Monotone Variational Inequalities

We now analyze the case where instead of having an accurate $F(\cdot)$, we only have access to a noisy estimator of $F(\cdot)$. This happens often in optimization and machine learning, where we sometimes do not have enough information or enough computational power to fully evaluate the operator $F(\cdot)$. Note that this is different from the additive noise $w_t$ in Algorithm 2 Line 4, which captures the stochasticity in choosing the FW direction.

In general, incorporating stochasticity into FW algorithms in optimization is known to be nontrivial when the variance of the noise, though bounded, is not sufficiently small. When there is only access to a noisy oracle for $F(x_t)$, the stochasticity enters the algorithm in a nonlinear manner through the computation of the smoothed FW direction. Consequently, developing algorithms with strong convergence guarantees becomes fundamentally more challenging. To illustrate this, suppose that we directly use a noisy estimate $F(x_t) + z_t$ (where $z_t$ represents the noise) in place of $F(x_t)$ in Algorithm 2 Line 3 to compute the FW direction $s_t$. Despite replacing the hardmin with a softmin (by introducing a regularizer), the FW direction $s_t$ remains highly sensitive to the noise $z_t$ because the FW direction computed from the exact $F(x_t)$ and its noisy counterpart $F(x_t) + z_t$ could differ significantly. As a result, due to the lack of control over the noise, it has been observed that using a noisy estimator of the operator $F(\cdot)$ in place of $F(\cdot)$ can lead to the divergence of the algorithm [16].

To overcome this issue, existing approaches to stochastic FW often build reduced-variance estimators of the operator $F(\cdot)$. One of the most common methods to achieve this is by averaging the estimates. Inspired by this approach, we develop a stochastic generalized FW algorithm for constrained MVI problems, where we first average the noisy estimates of $F(\cdot)$ through an iterative framework. This results in Algorithm 3 presented in the following. For ease of exposition, we only present the algorithm with constant stepsizes.

---

**Algorithm 3** Stochastic Frank-Wolfe for Monotone Variational Inequalities

---

1: **Input:** Integer $T$, tunable parameter $\tau$, and initialization $x_0 \in \mathcal{X}$ and $y_0 \in \mathbb{R}^d$.
2: **for** $t = 0, 1, \cdots, T - 1$ **do**
3: $\quad y_{t+1} = y_t + \beta(F(x_t) + z_t - y_t)$
4: $\quad s_t = \arg\min_{s \in \mathcal{X}}\{s^\top y_{t+1} + \tau f(s)\}$
5: $\quad x_{t+1} = x_t - \alpha(x_t - s_t)$
6: **end for**

---

The key step in Algorithm 3 is Line 3, where we build a sequence of estimators $\{y_t\}$ for $\{F(x_t)\}$ by averaging the newly observed noisy estimate $F(x_t) + z_t$ with past information. To illustrate, since $y_{t+1}$ is a convex combination (with parameter $\beta$) of the previous iterate $y_t$ and $F(x_t) + z_t$, we see that for any $t$, $y_t$ is essentially a convex combination (hence a weighted average) of $\{F(x_i) + z_i\}_{0 \le i \le t-1}$. Suppose that $x_t$ were stationary (i.e., $x_t \equiv x$ for some $x$), then $y_t$ effectively becomes a variance-reduced estimator of $F(x)$. To extend this idea to time-varying $x_t$, we choose the stepsizes $\alpha$ and $\beta$ such that $\beta \gg \alpha$, creating a two-timescale structure [9]. This ensures that, from the perspective of $y_t$, $x_t$ is nearly stationary, allowing $y_t$ to converge to $F(x_t)$ on a faster timescale. As a result, the $x_t$ iterates should behave similarly to the case where we have an accurate estimate of $F(\cdot)$. In Lemma 5.1, we show that this is indeed the case for the $y$-process.

To present the lemma, we first formally state our assumption on the noise sequence $\{z_t\}$.

**Assumption 5.1.** It holds for all $t \ge 0$ that $\mathbb{E}[z_t \mid \mathcal{F}_t] = 0$ and $\mathbb{E}[\|z_t\|_2^2 \mid \mathcal{F}_t] \le \sigma_z$ for some $\sigma_z > 0$, where $\mathcal{F}_t$ is the $\sigma$-algebra generated by $\{x_0, y_0, z_0, z_1, \cdots, z_{t-1}\}$.

The next lemma bounds the distance between the estimator $y_t$ and the desired target $F(x_t)$. The proof is presented in Appendix C.1.

**Lemma 5.1.** *Suppose that Assumptions 4.1 and 5.1 are satisfied and $\beta \in (0, 1)$. Then, we have for any $t \ge 0$ that*

$$\mathbb{E}[\|y_t - F(x_t)\|_2^2] \le \left(1 - \frac{3\beta}{4}\right)^t \|y_0 - F(x_0)\|^2 + \frac{8\beta\sigma_z}{3} + \frac{32 L_F D_{\mathcal{X}}^2 \alpha^2}{\beta^2}. \tag{7}$$

In view of the last term on the right-hand side of Eq. (7), to make $\mathbb{E}[\|y_t - F(x_t)\|_2^2]$ sufficiently small, we need the ratio between the stepsizes, i.e., $\alpha/\beta$, to be sufficiently small. This mathematically justifies the two-timescale structure in Algorithm 3. Notably, Lemma 5.1 holds irrespective of the monotonicity of $F(\cdot)$ and only makes use of its assumed Lipschitz continuity. Using this lemma allows us to prove the following result for stochastic FW in constrained MVI problems.

**Theorem 5.1.** *Consider $\{x_t\}$ generated by Algorithm 3. Suppose that $F(\cdot)$ is a monotone operator on $\mathcal{X}$, Assumptions 4.1, 4.2, and 5.1 are satisfied, and the regularizer $f(\cdot)$ satisfies Condition 4.1.*

*Then, when choosing $\beta = 8\alpha^{2/3}/3 \in (0,1)$, we have for any $t \geq 0$ that*

$$\mathbb{E}\left[\max_{s \in \mathcal{X}}(x_t - s)^\top F(x_t)\right] \leq \bar{c}_1 t(1-\alpha)^t + \frac{\bar{c}_2 \alpha^{1/3}}{\tau} + \frac{\bar{c}_3 \alpha^{2/3}}{\tau} + \frac{\bar{c}_4 \alpha}{\tau} + \bar{c}_5 \alpha + \tau \bar{f},$$

*where $\{\bar{c}_i\}_{1 \leq i \leq 5}$ are problem-dependent constants. See Appendix C.2 for their explicit expressions.*

A proof sketch of Theorem 5.1 is presented in Section 6, and the complete proof can be found in Appendix C.2. Based on Theorem 5.1, we have the following iteration complexity.

**Corollary 5.1.1.** *To achieve $\mathbb{E}[\max_{s \in \mathcal{X}}(x_t - s)^\top F(x_t)] \leq \epsilon$, the iteration complexity is $\tilde{\mathcal{O}}(\epsilon^{-6})$.*

Once again, we remark that, to the best of our knowledge, this appears to be the first algorithm with a last-iterate convergence guarantee in constrained stochastic MVI problems. The guarantees for variants of gradient-based algorithms are on the averaged iterate or under strong curvature assumptions on $F(\cdot)$ or $\mathcal{X}$ [8]. Numerical simulations are provided in Appendix D to verify the last-iterate convergence of our proposed algorithm.

Our iteration complexity of $\tilde{\mathcal{O}}(\epsilon^{-6})$ is slower than the $\mathcal{O}(\epsilon^{-2})$ enjoyed by stochastic mirror-prox algorithms in an averaged-iterate sense [36]. Despite the fact that we have last-iterate convergence, we believe that the above bound is not tight since using the same estimator in convex optimization problems results in an $\mathcal{O}(\epsilon^{-3})$ rate of convergence [45]. The reason for the potential looseness in our analysis is due to the fact that $F(\cdot)$ is not the gradient of a function and as such we must rely on the smoothness of the estimated FW direction $s_t$, which results in a suboptimal relationship between the hyperparameter $\tau$ and $\mathbb{E}[\|y_t - F(x_t)\|_2^2]$ in our analysis.

# 6 Proof Sketch of Theorem 5.1

Here, we present an outline of the proof of Theorem 5.1, which uses Lyapunov-based arguments. The proof of Theorem 4.1 follows a similar approach.

The first step in proving Theorem 5.1 is to establish Lemma 5.1, which shows that our constructed estimator $y_t$ indeed keeps track of the desired target $F(x_t)$.

## 6.1 Proof Sketch of Lemma 5.1

We will use $\|y_t - F(x_t)\|_2^2$ as a Lyapunov function to study the evolution of $y_t$. To begin with, for any $t \geq 0$, we have by Algorithm 3 Line 3 that

$$
\begin{aligned}
\|y_{t+1} - F(x_{t+1})\|_2^2 &= \|(1-\beta)y_t + \beta(F(x_t) + z_t) - F(x_{t+1})\|_2^2 \\
&= \|(1-\beta)(y_t - F(x_t)) + \beta z_t + F(x_t) - F(x_{t+1})\|_2^2 \\
&= (1-\beta)^2\|y_t - F(x_t)\|_2^2 + \beta^2\|z_t\|_2^2 + \|F(x_t) - F(x_{t+1})\|_2^2 \\
&\quad + 2(1-\beta)\beta(y_t - F(x_t))^\top z_t + 2\beta z_t^\top(F(x_t) - F(x_{t+1})) \\
&\quad + 2(1-\beta)(y_t - F(x_t))^\top(F(x_t) - F(x_{t+1})).
\end{aligned}
$$

Taking expectations conditioned on $\mathcal{F}_t$ on both sides of the previous inequality, and using Assumption 5.1 for the noise sequence $\{z_t\}$, along with some algebraic manipulations, we obtain

$$
\begin{aligned}
\mathbb{E}[\|y_{t+1} - F(x_{t+1})\|_2^2 \mid \mathcal{F}_t] &\leq \left(1 - \frac{3\beta}{4}\right)\|y_t - F(x_t)\|_2^2 + 2\beta^2\sigma_z \\
&\quad + \frac{6}{\beta}\mathbb{E}[\|F(x_t) - F(x_{t+1})\|_2^2 \mid \mathcal{F}_t].
\end{aligned}
$$

The details are presented in Appendix C.1. By the Lipschitz continuity of $F(\cdot)$ (cf. Assumption 4.1), we have

$$
\begin{aligned}
\mathbb{E}[\|F(x_t) - F(x_{t+1})\|_2^2 \mid \mathcal{F}_t] &\leq L_F \mathbb{E}[\|x_{t+1} - x_t\|_2^2 \mid \mathcal{F}_t] \\
&= L_F \alpha^2 \mathbb{E}[\|s_t - x_t\|_2^2 \mid \mathcal{F}_t] \\
&\leq 4L_F D_{\mathcal{X}}^2 \alpha^2,
\end{aligned}
$$

where the equality follows from Algorithm 3 Line 5. Combining the previous two inequalities together, we obtain

$$\mathbb{E}[\|y_{t+1} - F(x_{t+1})\|_2^2 \mid \mathcal{F}_t] \leq \left(1 - \frac{3\beta}{4}\right)\|y_t - F(x_t)\|_2^2 + 2\beta^2\sigma_z + \frac{24L_F D_{\mathcal{X}}^2\alpha^2}{\beta}.$$

The final result follows by first taking total expectations on both sides of the previous inequality and then repeatedly using the resulting inequality.

## 6.2 Proof Sketch of Theorem 5.1.

Given the proof of convergence for our estimator $y_t$ of $F(x_t)$, we now give an overview of the proof of Theorem 5.1. Let $s_t^* = \arg\min_{s \in \mathcal{X}}\{s^\top F(x_{t+1}) + \tau f(s)\}$. Using $V(\cdot)$ defined in Eq. (6) as our Lyapunov function, we have, after some algebra, that for any $t \geq 0$:

$$V(x_{t+1}) \leq (1 - \alpha)V(x_t) + \alpha F(x_t)^\top (s_t - s_{t-1}^*) + 2L_V D_{\mathcal{X}}^2\alpha^2, \tag{8}$$

where $L_V$ is a problem-dependent constant (cf. Lemma B.1). See Appendix C.2 for more details. It remains to bound the term $F(x_t)^\top (s_t - s_{t-1}^*)$. Using Lemma 4.1, we have

$$
\begin{aligned}
F(x_t)^\top (s_t - s_{t-1}^*) &\leq \|F(x_t)\|_2\|s_t - s_{t-1}^*\|_2 \\
&\leq \frac{\bar{F}L_F}{\tau\sigma_f}\|y_{t+1} - F(x_t)\|_2 \\
&\leq \frac{\bar{F}L_F}{\tau\sigma_f}(\|y_{t+1} - y_t\|_2 + \|y_t - F(x_t)\|_2) \\
&\leq \frac{\bar{F}L_F}{\tau\sigma_f}(\beta\|F(x_t) - y_t\|_2 + \beta\|z_t\|_2 + \|y_t - F(x_t)\|_2) \quad \text{(Algorithm 3 Line 3)} \\
&\leq \frac{\bar{F}L_F}{\tau\sigma_f}(\beta\|z_t\|_2 + 2\|y_t - F(x_t)\|_2),
\end{aligned}
$$

where the last inequality follows from $\beta \in (0,1)$. Using the upper bound we obtained for the term $F(x_t)^\top (s_t - s_{t-1}^*)$ in Eq. (8) and then taking total expectation on both sides of the resulting inequality, we have

$$\mathbb{E}[V(x_{t+1})] \leq (1 - \alpha)\mathbb{E}[V(x_t)] + \frac{\bar{F}L_F\beta\alpha}{\tau\sigma_f}\sigma_z^{1/2} + \frac{2\bar{F}L_F\alpha}{\tau\sigma_f}\mathbb{E}^{1/2}[\|y_t - F(x_t)\|_2^2] + 2L_V D_{\mathcal{X}}^2\alpha^2.$$

Finally, substituting the upper bound we obtained for $\mathbb{E}[\|F(x_{t+1}) - y_{t+1}\|_2^2]$ in Lemma 5.1 into the previous inequality and then repeatedly using it, we obtain the desired finite-time bound. The details are presented in Appendix C.2.

## 7 Conclusion

In this work, we study generalized FW algorithms in constrained MVI problems. We show that the algorithm enjoys last-iterate convergence guarantees. As a consequence of our results, we prove the rate of convergence for smoothed FP. We believe that this class of algorithms warrants further exploration. While extragradient and optimistic gradient algorithms can be seen as approximations to proximal point algorithms and consequently can be seen as discretizations of the ODE $\dot{x} = F(x)$, FW algorithms and consequently generalized FW algorithms are not necessarily related to this ODE and can thus be viewed as a new class of algorithms for such problems.

An interesting future direction of this work is to develop results for the case where $\tau = 0$. This corresponds to the regime in which FW algorithms are particularly useful since they require no projection and instead only an LMO. However, as commented on in [24], without additional curvature assumptions on $\mathcal{X}$ besides convexity, this case seems fundamentally more difficult due to the potential non-uniqueness and non-smoothness of the FW direction $s = \arg\min_{s \in \mathcal{X}} s^\top F(x)$.

## Acknowledgements

Z. Chen acknowledges support from the PIMCO Postdoctoral Fellowship. E. Mazumdar acknowledges support from NSF Award 2240110.

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

# Appendices

## A  Proof of Theorem 3.1

Observe that Algorithm 1 is a special case of Algorithm 2 with

$$F(x) = F(\pi^1, \pi^2) = \begin{bmatrix} 0 & -R \\ R^\top & 0 \end{bmatrix} \begin{bmatrix} \pi^1 \\ \pi^2 \end{bmatrix},$$

$$J(x) = \begin{bmatrix} 0 & -R \\ R^\top & 0 \end{bmatrix}^\top,$$

$$f(x) = f(\pi^1, \pi^2) = -\nu(\pi^1) - \nu(\pi^2) + \log(|\mathcal{A}^1||\mathcal{A}^2|),$$

$$w_t = \begin{bmatrix} e(A_t^1) - v_t^1 \\ e(A_t^2) - v_t^2 \end{bmatrix}.$$

In addition, since $\max_{a^1, a^2} |R(a^1, a^2)| \leq 1$ we have $L_F \leq \sqrt{|\mathcal{A}^1||\mathcal{A}^2|}$, $L_J = 0$, $\bar{F} \leq \sqrt{|\mathcal{A}^1| + |\mathcal{A}^2|}$, $\bar{f} \leq \log(|\mathcal{A}^1||\mathcal{A}^2|)$, $\sigma_f = 1$, $\sigma_w \leq 8$, and $D_{\mathcal{X}} \leq 2$. Now, applying Theorem 4.1 with $\alpha_t = 1/(t+1)$, since $\tau \leq 1$, we have

$$\mathrm{NG}(\pi^1, \pi^2) \leq \frac{4\sqrt{|\mathcal{A}^1| + |\mathcal{A}^2|}}{t+1} + \frac{36|\mathcal{A}^1||\mathcal{A}^2|}{\tau} \frac{\log(t+1)}{t+1} + \tau \log(|\mathcal{A}^1||\mathcal{A}^2|).$$

## B  Proof of All Technical Results in Section 4

### B.1  Proof of Lemma 4.1

Since $s^\top F(x) + \tau f(s)$ as a function of $s$ is $\tau\sigma_f$-strongly convex uniformly for all $x \in \mathcal{X}$ and the feasible set $\mathcal{X}$ is convex and compact, there is a unique global minimizer to the optimization problem $\min_{s \in \mathcal{X}}\{s^\top F(x) + \tau f(s)\}$. Moreover, since $f(\cdot)$ is chosen such that $\lim_{s \to \partial \mathcal{X}} \|\nabla f(s)\| = +\infty$ (cf. Condition 4.1), the unique optimal solution to $\min_{s \in \mathcal{X}}\{s^\top F(x) + \tau f(s)\}$ must lie in the interior of the feasible set $\mathcal{X}$. Therefore, for any $x_1, x_2 \in \mathcal{X}$, we have by the first-order optimality condition that

$$F(x_1) + \tau\nabla f(s(x_1)) = 0, \quad F(x_2) + \tau\nabla f(s(x_2)) = 0.$$

It follows that

$$F(x_1) - F(x_2) = \tau(\nabla f(s(x_2)) - \nabla f(s(x_1))).$$

Using the $\sigma_f$-strong convexity of $f(\cdot)$ and Assumption 4.1, we have

$$\begin{aligned} \tau\sigma_f \|s(x_1) - s(x_2)\|_2 &\leq \tau\|\nabla f(s(x_2)) - \nabla f(s(x_1))\|_2 \\ &= \|F(x_1) - F(x_2)\|_2 \\ &\leq L_F \|x_1 - x_2\|_2, \end{aligned}$$

which implies $\|s(x_1) - s(x_2)\|_2 \leq \frac{L_F}{\tau\sigma_f}\|x_1 - x_2\|_2$.

### B.2  Proof of Theorem 4.1

Recall that we use $V : \mathcal{X} \to \mathbb{R}$ defined as $V(x) = \max_{s \in \mathcal{X}}\{(x - s)^\top F(x) - \tau f(s)\}$ as our Lyapunov function. The goal here is to show that $x_t$ updated according to Algorithm 2 produces a negative drift with respect to $V(\cdot)$. The following lemma is needed to establish the result.

**Lemma B.1.** *The Lyapunov function $V(\cdot)$ is $L_V$-smooth with respect to $\|\cdot\|_2$, where $L_V = 2L_F + \frac{L_F^2}{\tau\sigma_f} + 2D_{\mathcal{X}}L_J$.*

*Proof of Lemma B.1.* To show the smoothness of $V(\cdot)$, it is enough to show that the gradient operator $\nabla V(\cdot)$ is Lipschitz continuous. To compute the gradient of $V(\cdot)$, apply Danskin's theorem [7] and we have

$$\nabla V(x) = F(x) + J(x)^\top (x - s(x)), \quad \forall x \in \mathcal{X},$$

where we recall that $s(x) = \arg\max_{s \in \mathcal{X}} \{(x - s)^\top F(x) - \tau f(s)\}$. For any $x_1, x_2 \in \mathcal{X}$, we have

$$
\begin{aligned}
\nabla V(x_1) - \nabla V(x_2) &= F(x_1) + J(x_1)^\top (x_1 - s(x_1)) - F(x_2) - J(x_2)^\top (x_2 - s(x_2)) \\
&= F(x_1) - F(x_2) + J(x_1)^\top (x_1 - x_2 + s(x_2) - s(x_1)) \\
&\quad + (J(x_1) - J(x_2))^\top (x_2 - s(x_2)).
\end{aligned}
$$

Using triangle inequality, Assumption 4.1, Assumption 4.2, and Lemma 4.1, we obtain

$$
\begin{aligned}
\|\nabla V(x_1) - \nabla V(x_2)\|_2 &\leq \|F(x_1) - F(x_2)\|_2 + \|J(x_1)\|_2 (\|x_1 - x_2\|_2 + \|s(x_2) - s(x_1)\|_2) \\
&\quad + \|J(x_1) - J(x_2)\|_2 \|x_2 - s(x_2)\|_2 \\
&\leq L_F \|x_1 - x_2\|_2 + L_F \left(1 + \frac{L_F}{\tau \sigma_f}\right) \|x_1 - x_2\|_2 + 2 D_{\mathcal{X}} L_J \|x_1 - x_2\|_2 \\
&= \left(2 L_F + \frac{L_F^2}{\tau \sigma_f} + 2 D_{\mathcal{X}} L_J\right) \|x_1 - x_2\|_2 \\
&= L_V \|x_1 - x_2\|_2.
\end{aligned}
$$

$\square$

Now, we are ready to prove Theorem 4.1. Using the smoothness of $V(\cdot)$, the explicit expression of $\nabla V(x_t)$ (both established in Lemma B.1), and the update equation in Algorithm 2 Line 3, we have for any $t \geq 0$ that

$$
\begin{aligned}
V(x_{t+1}) &\leq V(x_t) + \nabla V(x_t)^\top (x_{t+1} - x_t) + \frac{L_V}{2} \|x_{t+1} - x_t\|_2^2 \\
&= V(x_t) - \alpha_t (F(x_t) + J(x_t)^\top (x_t - s_t))^\top (x_t - s_t + w_t) + \frac{L_V \alpha_t^2}{2} \|x_t - s_t + w_t\|_2^2 \\
&= V(x_t) - \alpha_t F(x_t)^\top (x_t - s_t + w_t) - \alpha_t (x_t - s_t)^\top J(x_t)(x_t - s_t + w_t) \\
&\quad + \frac{L_V \alpha_t^2}{2} \|x_t - s_t + w_t\|_2^2.
\end{aligned}
$$

Taking expectations on both sides of the previous inequality, since $\{w_t\}$ is a martingale difference sequence, we have

$$
\begin{aligned}
\mathbb{E}[V(x_{t+1})] &\leq \mathbb{E}[V(x_t)] + \alpha_t \mathbb{E}[F(x_t)^\top (s_t - x_t)] - \alpha_t \mathbb{E}[(x_t - s_t)^\top J(x_t)(x_t - s_t)] \\
&\quad + \frac{L_V \alpha_t^2}{2} \mathbb{E}[\|x_t - s_t + w_t\|_2^2].
\end{aligned}
\tag{9}
$$

Next, we bound each term on the right-hand side of the previous inequality.

For the first term on the right-hand side of Eq. (9), observe that

$$
F(x_t)^\top (s_t - x_t) \leq -F(x_t)^\top (x_t - s_t) + \tau f(s_t) = -V(x_t).
$$

Therefore, we have

$$
\mathbb{E}[F(x_t)^\top (s_t - x_t)] \leq -\mathbb{E}[V(x_t)].
\tag{10}
$$

For the second term on the right-hand side of Eq. (9), since $F(\cdot)$ is monotone, the Jacobian matrix $J(\cdot)$ is positive semidefinite. Therefore, we have

$$
\mathbb{E}[(x_t - s_t)^\top J(x_t)(x_t - s_t)] \geq 0.
\tag{11}
$$

For the third term on the right-hand side of Eq. (9), using Assumption 4.3, we have

$$
\begin{aligned}
\mathbb{E}[\|x_t - s_t + w_t\|_2^2] &= \mathbb{E}[\|x_t - s_t\|_2^2 + \|w_t\|_2^2 + 2(x_t - s_t)^\top w_t] \\
&\leq 4 D_{\mathcal{X}}^2 + \sigma_w.
\end{aligned}
\tag{12}
$$

Combining Eqs. (10), (11), and (12) with Eq. (9), we have

$$
\mathbb{E}[V(x_{t+1})] \leq (1 - \alpha_t) \mathbb{E}[V(x_t)] + \frac{L_V (4 D_{\mathcal{X}}^2 + \sigma_w) \alpha_t^2}{2}
$$

$$= (1 - \alpha_t)\mathbb{E}[V(x_t)] + c_1\alpha_t^2,$$

where we denote $c_1 = L_V(4D_{\mathcal{X}}^2 + \sigma_w)/2$ for simplicity of notation. Repeatedly using the previous inequality, we have for all $t \geq 0$ that

$$\mathbb{E}[V(x_t)] \leq \prod_{j=1}^{t-1}(1 - \alpha_j)V(x_0) + c_1 \sum_{i=0}^{t-1}\alpha_i^2 \prod_{j=i+1}^{t-1}(1 - \alpha_j).$$

Finally, since

$$\max_{s \in \mathcal{X}}(x - s)^\top F(x) - \tau\bar{f} \leq V(x) \leq \max_s(x - s)^\top F(x) \leq 2D_{\mathcal{X}}\bar{F}, \quad \forall x \in \mathcal{X},$$

we have for all $t \geq 0$ that

$$\mathbb{E}\left[\max_{s \in \mathcal{X}}(x_t - s)^\top F(x_t)\right] \leq 2D_{\mathcal{X}}\bar{F}\prod_{j=1}^{t-1}(1 - \alpha_j) + c_1 \sum_{i=0}^{t-1}\alpha_i^2 \prod_{j=i+1}^{t-1}(1 - \alpha_j) + \tau\bar{f}. \quad (13)$$

**Constant Stepsizes.** When $\alpha_t \equiv \alpha$, we have

$$\prod_{j=0}^{t-1}(1 - \alpha_j) = (1 - \alpha)^t, \quad \text{and} \quad \sum_{i=0}^{t-1}\alpha_i^2 \prod_{j=i+1}^{t-1}(1 - \alpha_j) \leq \alpha.$$

It follows from Eq. (13) that

$$\mathbb{E}\left[\max_{s \in \mathcal{X}}(x_t - s)^\top F(x_t)\right] \leq 2D_{\mathcal{X}}\bar{F}(1 - \alpha)^t + c_1\alpha + \tau\bar{f}.$$

**Diminishing Stepsizes.** When $\alpha_t = 1/(t + 1)$ for all $t \geq 0$, we have

$$\prod_{j=1}^{t-1}(1 - \alpha_j) = \prod_{j=1}^{t-1}\frac{j}{j + 1} = \frac{1}{t + 1}$$

and

$$\sum_{i=0}^{t-1}\alpha_i^2 \prod_{j=i+1}^{t-1}(1 - \alpha_j) = \sum_{i=0}^{t-1}\frac{1}{(i + 1)^2}\prod_{j=i+1}^{t-1}\frac{j}{j + 1} = \frac{1}{t + 1}\sum_{i=0}^{t-1}\frac{1}{(i + 1)} \leq \frac{\log(t + 1)}{t + 1}.$$

It follows that

$$\mathbb{E}\left[\max_{s \in \mathcal{X}}(x_t - s)^\top F(x_t)\right] \leq \frac{2D_{\mathcal{X}}\bar{F}}{t + 1} + \frac{c_1\log(t + 1)}{t + 1} + \tau\bar{f}.$$

## C  Proof of All Technical Results in Section 5

### C.1  Proof of Lemma 5.1

For any $t \geq 0$, we have

$$
\begin{aligned}
\|y_{t+1} - F(x_{t+1})\|_2^2 &= \|(1 - \beta)y_t + \beta(F(x_t) + z_t) - F(x_{t+1})\|_2^2 \\
&= \|(1 - \beta)(y_t - F(x_t)) + \beta z_t + F(x_t) - F(x_{t+1})\|_2^2 \\
&= (1 - \beta)^2\|y_t - F(x_t)\|_2^2 + \beta^2\|z_t\|_2^2 + \|F(x_t) - F(x_{t+1})\|_2^2 \\
&\quad + 2(1 - \beta)\underbrace{(y_t - F(x_t))^\top(F(x_t) - F(x_{t+1}))}_{:=E_1} \\
&\quad + 2\beta\underbrace{z_t^\top(F(x_t) - F(x_{t+1}))}_{:=E_2} + 2(1 - \beta)\beta(y_t - F(x_t))^\top z_t. \quad (14)
\end{aligned}
$$

Using Cauchy–Schwarz inequality, we have

$$E_1 = (y_t - F(x_t))^\top(F(x_t) - F(x_{t+1}))$$

$$\leq \|y_t - F(x_t)\|_2 \|F(x_t) - F(x_{t+1})\|_2$$
$$\leq \frac{\beta}{8}\|y_t - F(x_t)\|_2^2 + \frac{2}{\beta}\|F(x_t) - F(x_{t+1})\|_2^2,$$

where the last inequality follows from $a^2 + b^2 \geq 2ab$ for any $a, b \in \mathbb{R}$. Similarly, we also have

$$E_2 = z_t^\top (F(x_t) - F(x_{t+1}))$$
$$\leq \|z_t\|_2 \|F(x_t) - F(x_{t+1})\|_2$$
$$\leq \frac{\beta}{2}\|z_t\|_2^2 + \frac{1}{2\beta}\|F(x_t) - F(x_{t+1})\|_2^2.$$

Combining the previous two inequalities with Eq. (14) and then taking expectations on both sides, we obtain

$$\mathbb{E}[\|y_{t+1} - F(x_{t+1})\|_2^2] \leq (1-\beta)^2 \mathbb{E}[\|y_t - F(x_t)\|_2^2] + \beta^2 \mathbb{E}[\|z_t\|_2^2] + \mathbb{E}[\|F(x_t) - F(x_{t+1})\|_2^2]$$
$$+ \frac{\beta}{4}\mathbb{E}[\|y_t - F(x_t)\|_2^2] + \frac{4}{\beta}\mathbb{E}[\|F(x_t) - F(x_{t+1})\|_2^2]$$
$$+ \beta^2 \mathbb{E}[\|z_t\|_2^2] + \mathbb{E}[\|F(x_t) - F(x_{t+1})\|_2^2]$$
$$+ 2(1-\beta)\beta \mathbb{E}[(y_t - F(x_t))^\top \mathbb{E}[z_t \mid \mathcal{F}_t]]$$
$$\leq \left(1 - \frac{3\beta}{4}\right)\mathbb{E}[\|y_t - F(x_t)\|_2^2] + 2\beta^2 \sigma_z + \frac{6}{\beta}\underbrace{\mathbb{E}[\|F(x_t) - F(x_{t+1})\|_2^2]}_{:=E_3},$$

where the last line follows from Assumption 5.1 and the fact that $\beta \in (0,1)$.

It remains to bound the term $E_3$ on the right-hand side of the previous inequality. Observe that

$$\mathbb{E}[\|F(x_t) - F(x_{t+1})\|_2^2] \leq L_F \mathbb{E}[\|x_{t+1} - x_t\|_2^2] \qquad \text{(Assumption 4.1)}$$
$$= L_F \alpha^2 \mathbb{E}[\|s_t - x_t\|_2^2] \qquad \text{(Algorithm 3 Line 5)}$$
$$\leq 4L_F D_{\mathcal{X}}^2 \alpha^2.$$

Therefore, we have

$$\mathbb{E}[\|y_{t+1} - F(x_{t+1})\|_2^2] \leq \left(1 - \frac{3\beta}{4}\right)\mathbb{E}[\|y_t - F(x_t)\|_2^2] + 2\beta^2 \sigma_z + \frac{24L_F D_{\mathcal{X}}^2 \alpha^2}{\beta}.$$

Repeatedly using the previous inequality, we obtain

$$\mathbb{E}[\|y_t - F(x_t)\|_2^2] \leq \left(1 - \frac{3\beta}{4}\right)^t \|y_0 - F(x_0)\|_2^2 + \frac{8\beta\sigma_z}{3} + \frac{32L_F D_{\mathcal{X}}^2 \alpha^2}{\beta^2}.$$

### C.2 Proof of Theorem 5.1

Recall that $s_t^* := \arg\min_{s \in \mathcal{X}}\{s^\top F(x_{t+1}) + \tau f(s)\}$. Using the smoothness of $V(\cdot)$, the explicit expression of $\nabla V(x_t)$ (both established in Lemma B.1), and the update equation in Algorithm 3 Line 5, we have for any $t \geq 0$ that

$$V(x_{t+1}) \leq V(x_t) + \nabla V(x_t)^\top (x_{t+1} - x_t) + \frac{L_V}{2}\|x_{t+1} - x_t\|_2^2$$
$$= V(x_t) - \alpha(F(x_t) + J(x_t)^\top (x_t - s_t))^\top (x_t - s_t) + \frac{L_V \alpha^2}{2}\|x_t - s_t\|_2^2$$
$$= V(x_t) - \alpha F(x_t)^\top (x_t - s_t) - \alpha(x_t - s_t)^\top J(x_t)(x_t - s_t) + \frac{L_V \alpha^2}{2}\|x_t - s_t\|_2^2$$
$$\leq V(x_t) - \alpha F(x_t)^\top (x_t - s_t) + 2L_V D_{\mathcal{X}}^2 \alpha^2 \qquad \text{($J(\cdot)$ is positive semidefinite)}$$
$$= V(x_t) - \alpha F(x_t)^\top (x_t - s_{t-1}^*) - \alpha F(x_t)^\top (s_{t-1}^* - s_t) + 2L_V D_{\mathcal{X}}^2 \alpha^2$$
$$\leq (1-\alpha)V(x_t) + \alpha F(x_t)^\top (s_t - s_{t-1}^*) + 2L_V D_{\mathcal{X}}^2 \alpha^2, \qquad (15)$$

where the last line follows from the definition of $V(\cdot)$. To proceed, observe that

$$F(x_t)^\top (s_t - s_{t-1}^*) \leq \|F(x_t)\|_2 \|s_t - s_{t-1}^*\|_2$$

$$\leq \frac{\bar{F}L_F}{\tau\sigma_f}\|y_{t+1} - F(x_t)\|_2 \qquad\qquad\qquad \text{(Lemma 4.1)}$$

$$\leq \frac{\bar{F}L_F}{\tau\sigma_f}(\|y_{t+1} - y_t\|_2 + \|y_t - F(x_t)\|_2)$$

$$\leq \frac{\bar{F}L_F}{\tau\sigma_f}(\beta\|F(x_t) - y_t\|_2 + \beta\|z_t\|_2 + \|y_t - F(x_t)\|_2) \quad \text{(Algorithm 3 Line 3)}$$

$$\leq \frac{\bar{F}L_F}{\tau\sigma_f}(\beta\|z_t\|_2 + 2\|y_t - F(x_t)\|_2),$$

where the last line follows from $\beta \in (0,1)$. Combining the previous inequality with Eq. (15), we have

$$V(x_{t+1}) \leq (1-\alpha)V(x_t) + \frac{\bar{F}L_F\beta\alpha}{\tau\sigma_f}\|z_t\|_2 + \frac{2\bar{F}L_F\alpha}{\tau\sigma_f}\|y_t - F(x_t)\|_2 + 2L_V D_{\mathcal{X}}^2\alpha^2.$$

Taking expectations on both sides of the previous inequality, we have

$$\mathbb{E}[V(x_{t+1})] \leq (1-\alpha)\mathbb{E}[V(x_t)] + \frac{\bar{F}L_F\beta\alpha}{\tau\sigma_f}\sigma_z^{1/2} + \frac{2\bar{F}L_F\alpha}{\tau\sigma_f}\mathbb{E}^{1/2}[\|y_t - F(x_t)\|_2^2] + 2L_V D_{\mathcal{X}}^2\alpha^2,$$

where we used $\mathbb{E}[\|z_t\|_2] \leq \mathbb{E}^{1/2}[\|z_t\|_2^2] = \mathbb{E}^{1/2}[\mathbb{E}[\|z_t\|_2^2 \mid \mathcal{F}_t]] \leq \sigma_z^{1/2}$ and $\mathbb{E}[\|y_t - F(x_t)\|_2] \leq \mathbb{E}^{1/2}[\|y_t - F(x_t)\|_2^2]$. Using Lemma 5.1 to bound $\mathbb{E}[\|y_t - F(x_t)\|_2^2]$, we have

$$\mathbb{E}[V(x_{t+1})] \leq (1-\alpha)\mathbb{E}[V(x_t)] + \frac{\bar{F}L_F\beta\alpha}{\tau\sigma_f}\sigma_z^{1/2} + 2L_V D_{\mathcal{X}}^2\alpha^2$$
$$+ \frac{2\bar{F}L_F\alpha}{\tau\sigma_f}\left(\left(1 - \frac{3\beta}{4}\right)^{t/2}\|y_0 - F(x_0)\|_2 + 3\beta^{1/2}\sigma_z^{1/2} + \frac{6L_F^{1/2}D_{\mathcal{X}}\alpha}{\beta}\right),$$

where we use $\sqrt{a+b+c} \leq \sqrt{a} + \sqrt{b} + \sqrt{c}$ for any $a,b,c \geq 0$. Repeatedly using the previous inequality, since choosing $\beta = 8\alpha^{2/3}/3$ implies $1 - 3\beta/4 \leq (1-\alpha)^2$, we have

$$\mathbb{E}[V(x_t)] \leq (1-\alpha)^t V(x_0) + \frac{\bar{F}L_F\beta}{\tau\sigma_f}\sigma_z^{1/2} + \frac{2\bar{F}L_F\alpha}{\tau\sigma_f}t(1-\alpha)^{t-1}\|y_0 - F(x_0)\|_2$$
$$+ \frac{2\bar{F}L_F}{\tau\sigma_f}\left(3\beta^{1/2}\sigma_z^{1/2} + \frac{6L_F^{1/2}D_{\mathcal{X}}\alpha}{\beta}\right) + 2L_V D_{\mathcal{X}}^2\alpha.$$

Using $\beta = 8\alpha^{2/3}/3 \in (0,1)$ (which also implies $\alpha \leq 1 - \alpha$) and the explicit expression of $L_V$ from Lemma B.1 in the previous inequality, we have

$$\mathbb{E}[V(x_t)] \leq (1-\alpha)^t V(x_0) + \frac{8\bar{F}L_F\sigma_z^{1/2}\alpha^{2/3}}{3\tau\sigma_f} + \frac{2\bar{F}L_F}{\tau\sigma_f}t(1-\alpha)^t\|y_0 - F(x_0)\|_2$$
$$+ \frac{18\bar{F}L_F}{\tau\sigma_f}\left(\sigma_z^{1/2} + \frac{L_F^{1/2}D_{\mathcal{X}}}{4}\right)\alpha^{1/3} + 4(L_F + D_{\mathcal{X}}L_J)D_{\mathcal{X}}^2\alpha + \frac{2L_F^2 D_{\mathcal{X}}^2}{\tau\sigma_f}\alpha$$

Recall that

$$\max_{s\in\mathcal{X}}(x-s)^\top F(x) - \tau\bar{f} \leq V(x) \leq \max_s(x-s)^\top F(x) \leq 2D_{\mathcal{X}}\bar{F}, \quad \forall x \in \mathcal{X}.$$

Therefore, we obtain the following finite-time bound:

$$\mathbb{E}\left[\max_{s\in\mathcal{X}}(x_t - s)^\top F(x_t)\right]$$

$$\leq 2D_{\mathcal{X}}\bar{F}(1-\alpha)^t + \frac{8\bar{F}L_F\sigma_z^{1/2}\alpha^{2/3}}{3\tau\sigma_f} + \frac{2\bar{F}L_F}{\tau\sigma_f}t(1-\alpha)^t\|y_0 - F(x_0)\|_2$$

$$+ \frac{18\bar{F}L_F}{\tau\sigma_f}\left(\sigma_z^{1/2} + \frac{L_F^{1/2}D_{\mathcal{X}}}{4}\right)\alpha^{1/3} + 4(L_F + D_{\mathcal{X}}L_J)D_{\mathcal{X}}^2\alpha + \frac{2L_F^2 D_{\mathcal{X}}^2}{\tau\sigma_f}\alpha + \tau\bar{f}$$

$$= \bar{c}_1 t \left(1 - \alpha\right)^t + \frac{\bar{c}_2 \alpha^{1/3}}{\tau} + \frac{\bar{c}_3 \alpha^{2/3}}{\tau} + \frac{\bar{c}_4 \alpha}{\tau} + \bar{c}_5 \alpha + \tau \bar{f},$$

where

$$\bar{c}_1 = 2 D_{\mathcal{X}} \bar{F} + \frac{2 \bar{F} L_F}{\tau \sigma_f} \| y_0 - F(x_0) \|_2, \ \bar{c}_2 = \frac{18 \bar{F} L_F}{\sigma_f} \left( \sigma_z^{1/2} + \frac{L_F^{1/2} D_{\mathcal{X}}}{4} \right), \ \bar{c}_3 = \frac{8 \bar{F} L_F \sigma_z^{1/2}}{3 \sigma_f},$$

$$\bar{c}_4 = \frac{2 D_{\mathcal{X}}^2 L_F^2}{\sigma_f}, \ \bar{c}_5 = 4(L_F + D_{\mathcal{X}} L_J) D_{\mathcal{X}}^2.$$

The proof is complete.

## D  Numerical Simulations

In this section, we conduct numerical simulations to empirically verify the performance of our proposed algorithms.

### D.1  Generalized Frank-Wolfe

We compare the performance of Algorithm 2 with the extragradient method, both of which provably enjoy an $\mathcal{O}(T^{-1/2})$ rate of last-iterate convergence.

Recall that the MVI problem aims to find an $x^* \in \mathbb{R}^d$ such that $\max_{s \in \mathcal{X}} (x^* - s)^\top F(x^*) \leq 0$. According to [15], the extragradient algorithm initializes an $x_0 \in \mathcal{X}$ and update $x_k$ iteratively according to the following formula:

$$x_{t+1/2} = \Pi_{\mathcal{X}}(x_t - \alpha F(x_t)), \quad x_{t+1} = \Pi_{\mathcal{X}}(x_t - \alpha F(x_{t+1/2})),$$

where $\Pi_{\mathcal{X}}(\cdot)$ the projection operator onto $\mathcal{X}$ with respect to $\| \cdot \|_2$.

#### D.1.1  Rock-Paper-Scissors Game

The Rock-Paper-Scissors game is a classic example of a zero-sum game, where each player has three actions: Rock, Paper, or Scissors. The rules are such that Rock beats Scissors, Scissors beat Paper, and Paper beats Rock. If both players choose the same move, the game results in a tie. The payoff matrix for the player 1 (i.e., the row player) can be represented as follows:

|  | Rock | Paper | Scissors |
|---|---|---|---|
| Rock | 0 | −1 | 1 |
| Paper | 1 | 0 | −1 |
| Scissors | −1 | 1 | 0 |

The results for implementing generalized FW and the extragradient method are reported in Figure 1.

#### D.1.2  The Burglar-Policeman Matrix Game

The Burglar-Policeman matrix game is another classic zero-sum game. The burglar wants to avoid being caught, while the policeman wants to catch the burglar. In this game, the actions available to both players are to either "Stay" at their current position or "Switch" to another location. The payoff matrix for the burglar can be represented as follows:

|  | Policeman Stay | Policeman Switch |
|---|---|---|
| Burglar Stay | −1 | 1 |
| Burglar Switch | 1 | −1 |

The results for implementing generalized FW and the extragradient method are reported in Figure 2.

In either the Rock-Paper-Scissors Game or the Burglar-Policeman Matrix Game, both algorithms seem to have similar performance. However, the extragradient method seems to be more stable.

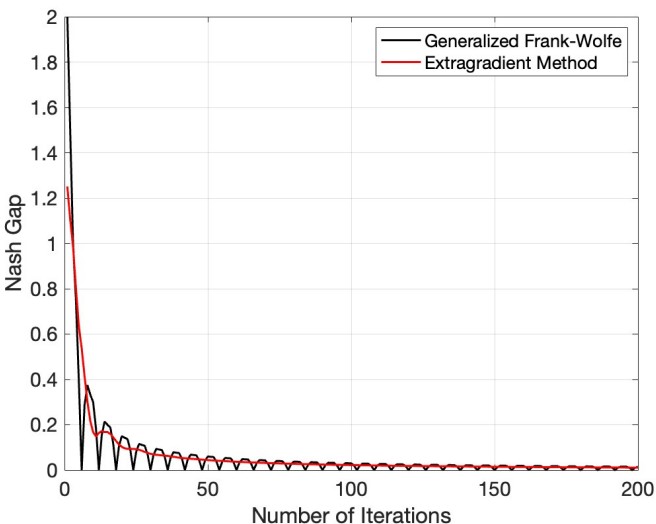

Figure 1: Convergence Rate Comparison for the Rock-Paper-Scissors Game

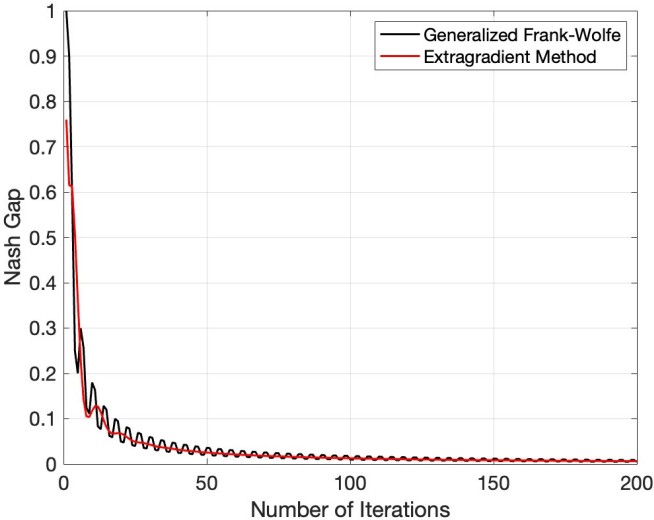

Figure 2: Convergence Rate Comparison for the Burglar-Policeman Matrix Game

### D.1.3 A Randomly Generated Matrix Game

In this experiment, we choose $F(x) = Mx$ and $\mathcal{X} = \Delta^{100} \times \Delta^{100}$, where $\Delta^{100}$ denotes the 100-dimensional probability simplex. The matrix $M$ is chosen to be $M = [\mathbf{0}^{100 \times 100}, -R; R^\top, \mathbf{0}^{100 \times 100}]$, where $R \in \mathbb{R}^{100 \times 100}$ is a randomly generated matrix. The MVI problem can alternatively be interpreted as a zero-sum game with the payoff matrix being $R$. The results are reported in Figure 3.

From the numerical simulations, we see that the generalized FW algorithm seems to slightly outperform the extragradient algorithm in the beginning. However, asymptotically, the extragradient method seems to perform better. This makes intuitive sense as using the softmax in the generalized FW algorithms results in a bias that depends on $\tau$. The numerical simulations verify that the generalized FW algorithm indeed holds practical potential.

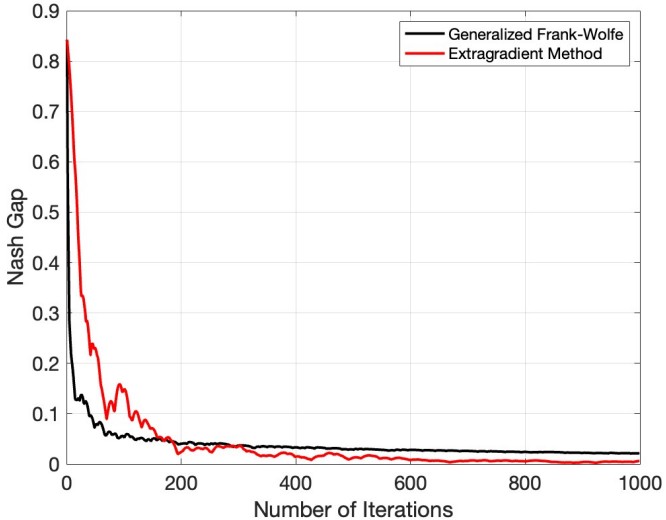

Figure 3: Convergence Rate Comparison for

## D.2 Stochastic Generalized Frank-Wolfe

Since none of the existing algorithms (e.g., the extragradient method, the optimistic gradient method, and the Halpern iteration method) have provable convergence in the stochastic setting, we will only verify the convergence of our algorithm here. The experiment setup is the same as in Appendix D.1.3 except that $F(x)$ is replaced by $F(x) + z$, where $z$ is a bounded random variable.

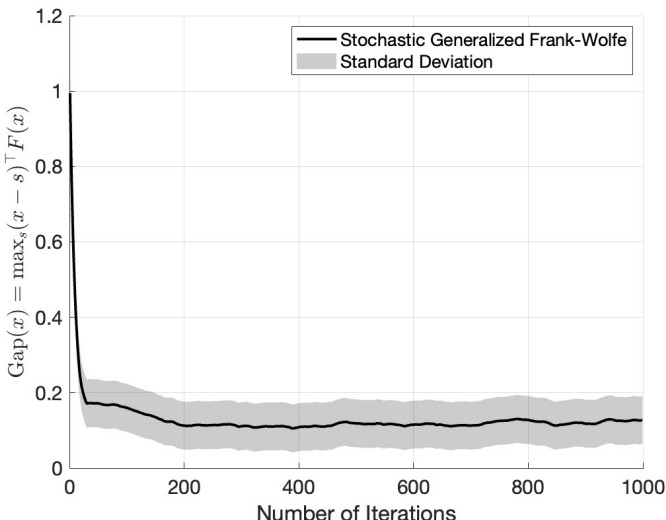

Figure 4: Convergence of Algorithm 3

In the stochastic setting, we choose $\tau = 0.1$, $\alpha = 0.1$, and $\beta = 0.01$ in Algorithm 3. The result is reported in Figure 4. From Figure 4, we see that Algorithm 3 indeed converges, but not to zero due to the stochasticity and the fact that we are using constant stepsizes, which agrees with Theorem 5.1. The fact that our algorithm is stable in the stochastic setting highlights one of its main advantages compared with the existing methods.

