# OpenReview forum: "Last-Iterate Convergence for Generalized Frank-Wolfe in Monotone Variational Inequalities"
_NeurIPS.cc/2024/Conference — NeurIPS 2024 poster_

### Official Review · Reviewer_3YVN · 2024-06-29

**Soundness:** 2
**Presentation:** 3
**Contribution:** 2
**Rating:** 5
**Confidence:** 4

**Summary:**

The paper presents a regularized version of Frank-Wolfe algorithm for monotone variational inequalities for which the authors an $\tilde{O}(T^{-1/2})$ convergence rate of the last iterate. In the stochastic case, using the variance reduction technique, the authors show an algorithm that enjoys $\tilde{O}(T^{-1/6})$ last-iterate convergence.

**Strengths:**

- The presentation and the analysis of the paper are mostly straightforward and easy to follow.
- Using regularization and variance reduction is a simple idea but it appears to work quite well.

**Weaknesses:**

1. There are some major issues with the proofs in the paper, as follows
- In Line 808, second equation, $w_t$ does not appear. How would the authors fix this issue? It seems non-trivial to me, although in the easier case when $w_t=0$, the proof will work.
- Lemma 4.1: The optimality condition here is usually considered a very strong, ie, there exists a point such that the gradient = 0, especially when the function considered changes according to $x$.
- The assumption that there exists $\bar{F} = \max ||F(x)||_2$ seems unusual and strong and the dependence of the convergence on $\bar{F}$ is not convincing to me.
2. Empirically, there is a gap between extragradient and the proposed algorithm.

**Questions:**

Could the authors address my questions mentioned above? I would be happy to increase my score if these concerns are addressed adequately.

**Limitations:**

Yes

---

> ### Author Rebuttal · Authors · 2024-08-06
>
> >Comment: In Line 808 second equation, $w_t$ does not appear. How would the authors fix this issue? It seems non-trivial to me, although in the easier case when $w_t=0$, the proof will work.
>
> **Response:** We greatly appreciate the reviewer for carefully reading our work. The missing $w_t$ in Line 808 was indeed a typo. Next, we present a fix to this issue. We start with the decomposition of the term $E_1$ in Line 808:
>
> $$E_1=(s_t-s_{t+1})^\top (F(x_{t+1})-F(x_t))+(x_t-s_t)^\top (F(x_{t+1})-F(x_t))\quad\qquad (1)$$
>
> For the first term on the right-hand side of the previous inequality, we have by Assumption 4.1 and Lemma 4.1 that
> $$
> (s_t-s_{t+1})^\top (F(x_{t+1})-F(x_t))\leq \vert|s_t-s_{t+1}\vert|\vert|F(x_{t+1})-F(x_t)\vert|
> \leq \frac{L^2}{\tau \sigma_f}\vert|x_{t+1}-x_t\vert|^2.\quad (2)
> $$
> This is the same as what we did in Line 808 (from the first inequality to the third inequality).
>
> The typo was with our approach in bounding the second term on the right-hand side of Eq. (1). Next, we present a different approach to bound it, which requires the following additional but mild assumption.
>
> **Assumption:** The Jacobian matrix $J(\cdot)$ of the operator $F(\cdot)$ is Lipschitz continuous.
>
> Since $F(\cdot)$ is Lipschitz continuous (Assumption 4.1), its Jacobian matrix is well-defined almost everywhere. Assumption imposes a mild smoothness condition on the Jacobian matrix, which is automatically satisfied in the zero-sum game setting, where the operator $F(\cdot)$ is a linear operator. Note that the monotonicity of $F(x)$ implies that $J(x)+J(x)^\top$ is a positively semidefinite matrix for any $x\in\mathcal{X}$.
>
>
> Denote the Lipschitz constant of $J(\cdot)$ as $L_J$. To proceed, in view of the second term on the right-hand side of Eq. (1), for a given $v\in\mathbb{R}^d$, consider the function $g_v(x)=v^\top F(x)$. Since
> $$\vert|\nabla g_v(x_1)-\nabla g_v(x_2)\vert|=\vert|(J(x_1)-J(x_1))^\top v\vert|\leq L_J\vert|v\vert|\vert|x_1-x_2\vert|,$$
> the function $g_v(x)$ is an $L_J\vert|v\vert|$-smooth function with respect to $x$. Therefore, using the descent lemma, we have for any $x_1,x_2\in\mathcal{X}$:
> $$g_v(x_2)-g_v(x_1)\leq (x_2-x_1)^\top J(x_1)^\top v+\frac{L_J\vert|v\vert|}{2}\vert|x_1-x_2\vert|^2.$$
> Identifying $v=x_t-s_t$, $x_1=x_t$, and $x_2=x_{t+1}$ in the above inequality, the second term on the right-hand side of Eq. (1) can be bounded as follows:
> $$(x_t-s_t)^\top (F(x_{t+1})-F(x_t))\qquad\qquad\qquad\qquad\qquad\qquad\qquad\qquad\qquad\qquad\qquad$$
> $$\leq (x_{t+1}-x_t)J(x_t)^\top (x_t-s_t)+\frac{L_J\vert|x_t-s_t\vert|\vert|x_{t+1}-x_t\vert|^2}{2}\qquad\qquad\qquad\qquad\qquad\qquad\qquad$$
> $$= -\alpha_t(x_t-s_t+w_t)J(x_t)^\top (x_t-s_t)+\frac{L_J\vert|x_t-s_t\vert|\vert|x_{t+1}-x_t\vert|^2}{2}\qquad\qquad\qquad\qquad\qquad$$
> $$\leq -\alpha_t(x_t-s_t)^\top \left[\frac{J(x_t)+J(x_t)^\top}{2}\right] (x_t-s_t)-\alpha_tw_t^\top J(x_t)^\top (x_t-s_t)+\frac{2L_JD_{\mathcal{X}}}{2}\vert|x_{t+1}-x_t\vert|^2$$
> $$\leq -\alpha_tw_t^\top J(x_t)^\top (x_t-s_t)+\frac{2L_JD_{\mathcal{X}}}{2}\vert|x_{t+1}-x_t\vert|^2\qquad\qquad\qquad\qquad\qquad\qquad\qquad\qquad(3)
> $$
> where the first equality follows from the update equation for $x_t$, the second inequality follows from $\vert|x_t-s_t\vert|\leq \vert|x_t\vert|+\vert|s_t\vert|\leq 2D_{\mathcal{X}}$ (recall that $D_{\mathcal{X}}$ is the radius of the feasible set), and the last inequality follows from $J(x_t)+J(x_t)^\top$ being a positively semidefinite matrix (due to the monotonicity of $F(\cdot)$).

---

> ### Author Response · Authors · 2024-08-06
> **Rebuttal (Continued)**
>
> Combining Eqs. (2) and (3) with Eq. (1), assuming without loss of generality that the regularization parameter $\tau$ is chosen to be less than $1$, we have
> $$
> E_1=(s_t-s_{t+1})^\top (F(x_{t+1})-F(x_t))+(x_t-s_t)^\top (F(x_{t+1})-F(x_t))$$
> $$\leq -\alpha_tw_t^\top J(x_t)^\top (x_t-s_t)+\left(\frac{2L_JD_{\mathcal{X}}}{2}+\frac{L^2}{\tau \sigma_f}\right)\vert|x_{t+1}-x_t\vert|^2$$
> $$\leq -\alpha_tw_t^\top J(x_t)^\top (x_t-s_t)+\left(\frac{2L_JD_{\mathcal{X}}}{2}+\frac{L^2}{\sigma_f}\right)\frac{1}{\tau}\vert|x_{t+1}-x_t\vert|^2$$
> $$= -\alpha_tw_t^\top J(x_t)^\top (x_t-s_t)+ \frac{M}{\tau}\vert|x_{t+1}-x_t\vert|^2,\qquad\qquad\qquad(4)
> $$
> where we denote $M=\frac{2L_JD_{\mathcal{X}}}{2}+\frac{L^2}{\sigma_f}$ in Eq. (4) for simplcity of notation. To proceed, using the same line of analysis as we did from the third inequality to the last inequality in Line 808, we have
> $$
> E_1
> \leq -\alpha_tw_t^\top J(x_t)^\top (x_t-s_t)+\frac{M}{\tau}\vert|x_{t+1}-x_t\vert|^2\qquad\qquad\qquad$$
> $$\leq -\alpha_tw_t^\top J(x_t)^\top (x_t-s_t)+\frac{8\alpha_t^2MD_{\mathcal{X}}^2}{\tau} +\frac{2M\alpha_t^2}{\tau}\vert|w_t\vert|^2.
> $$
> In view of the previous inequality, the first term on the right-hand side is mean zero due to $\{w_t\}$ being a martingale difference sequence (cf. Assumption 4.2), and the rest of the terms are the same (up to a multiplicative constant that is independent of $\tau$) as the terms on the right-hand side of the last inequality in Line 808. Following the same analysis starting from Line 809, the overall finite-time bound (with the same rate of convergence) can be reproduced.
>
> **In summary**, the convergence analysis for Algorithm 2 can be reproduced with the additional assumption that the Jacobian matrix of $F(\cdot)$ is Lipschitz continuous. The analysis of Algorithm 1 for zero-sum games remains intact since the operator $F(\cdot)$ in that case is linear, hence the Jacobian is a constant, which is automatically Lipschitz. The analysis for Algorithm 3 remains intact as it does not require the additional assumption.
>
> It is an interesting future direction to investigate whether the additional Lipschitz continuity assumption on the Jacobian matrix is necessary for the convergence of Algorithm 2. Intuitively, when there is stochasticity in the algorithm, we need some form of smoothness in the update to control the stochastic error. For Algorithm 3, this is achieved through the algorithm design (by creating a timescale separation between the variance reduced stochastic estimate $y_t$ and the main iterate $x_t$). In Algorithm 2, since the noise is directly added to the update equation, we imposed the additional Lipschitz continuity assumption for the analysis.

---

> ### Author Response · Authors · 2024-08-06
> **Rebuttal (Continued)**
>
> >Comment: Lemma 4.1: The optimality condition here is usually considered very strong, ie, there exists a point such that the gradient = 0, especially when the function considered changes according to $x$.
>
> **Response:** To clarify, the optimality condition is actually a natural consequence of our analysis rather than an assumption. To elaborate, note that the function $s^\top F(x)+\tau f(s)$ as a function of $s$ is $\tau \sigma_f$-strongly convex uniformly for all $x\in\mathcal{X}$. The strongly convex property comes from the regularizer $\tau f(s)$ and is independent of where $x$ is. Therefore, since the optimization problem $\arg\min_{s\in\mathcal{X}}\{s^\top F(x)+\tau f(s)\}$ of finding the Frank-Wolfe direction is a strongly convex problem on a convex and compact feasible set, there is a unique global minimizer. Moreover, since $f(\cdot)$ is chosen such that $\lim_{s\rightarrow\partial \mathcal{X}}\vert|\nabla f(s)\vert|=+\infty$ (cf. Condition 4.1), the optimal solution of $\arg\min_{s\in\mathcal{X}}\{s^\top F(x)+\tau f(s)\}$ must lie in the interior of the feasible set $\mathcal{X}$ (otherwise we can easily argue a contradiction), which leads to the first-order optimality condition stated in the beginning of the proof of Lemma 4.1. In the next version, we will provide a more detailed illustration in the proof of Lemma 4.1 to support stating the optimality conditions this way.
>
>
> >Comment: The assumption that there exists $\bar{F}=\max\vert|F(x)\vert|_2$ seems unusual and strong and the dependence of the convergence on $\bar{F}$ is not convincing to me.
>
> **Response:** To clarify, this is also not an assumption but a natural consequence of our analysis. Since the feasible set $\mathcal{X}$ is compact, and the function $F(\cdot)$ is Lipschitz continuous (hence continuous), the quantity $\bar{F}:=\max_{x\in\mathcal{X}}\vert|F(x)\vert|$ is well-defined and finite according to the extreme value theorem. We will clarify this point in the next version of this work. In our convergence bound, $\bar{F}$ only appears as a constant, hence does not impact the overall convergence rate.
>
> >Comment: Empirically, there is a gap between extragradient and the proposed algorithm.
>
> **Response:** In Figure 1, it is not completely clear which algorithm has the better rate of convergence as Frank-Wolfe seems to decay faster in the beginning while EG has a better convergence rate afterward. Theoretically, both generalized Frank-Wolfe and EG have a comparable $O(1/\sqrt{T})$ rate of convergence. In the stochastic setting, EG fails to converge while our variant of Frank-Wolfe (i.e., Algorithm 3) has provable last-iterate convergence.
>
> That being said, the goal of this paper was not to propose an algorithm that outperforms existing benchmarks. In recent years, existing studies of monotone variational inequalities have primarily focused on gradient-based algorithms such as EG and OG. On the other hand, Frank-Wolfe is also a natural algorithm (but is, to some extent, overlooked), and has a nice connection with fictitious play in zero-sum games. We study this algorithm and show that with a simple modification (that is, adding a regularizer), the convergence rate of Frank-Wolfe matches with that of EG and OG in the deterministic setting and is provably convergent in the stochastic setting.

---

> ### Author Response · Authors · 2024-08-09
> **Rebuttal Feedback**
>
> We greatly appreciate the reviewer’s time and effort in evaluating our work. Please let us know if our response addresses the concerns raised, or if further clarification is needed.

---

> ### Comment · Reviewer_3YVN · 2024-08-09
>
> I thank the authors for the response. The authors have addressed some parts of my concerns, I have raised the score accordingly. However, I also think that the paper needs extra work to address the proof gap and the new assumption as well as justifications for existing condition/assumptions.

---

> ### Author Response · Authors · 2024-08-12
> **Follow-Up Comments by Authors**
>
> We are glad to hear that our response addresses some of the reviewer's concerns, and we greatly appreciate the reviewer for raising the score.
>
> The new proof has been presented in full detail in the previous response, and it will be included in the next version of this work. Next, we would like to provide justification for our newly imposed assumption, which does not undermine the contributions of this work. First, it does not imply any strong curvature properties (such as strong monotonicity of the operator or strong convexity of the feasible set). Moreover, our original motivation for allowing the update to be noisy in Algorithm 2 is that it can directly be used to obtain the convergence bound of smoothed fictitious play, where the newly imposed assumption is automatically satisfied due to the linear structure of the operator $F(\cdot)$. In either the completely deterministic setting (where $w_t=0$) or the more challenging stochastic setting (where $F(x_t)$ is noisy), both our original approaches work.

---

### Official Review · Reviewer_UPhr · 2024-07-07

**Soundness:** 3
**Presentation:** 3
**Contribution:** 3
**Rating:** 6
**Confidence:** 4

**Summary:**

This paper focuses on solving the monotone MVI. To address this problem, the authors propose a new algorithm, which is a generalized variant of the Frank-Wolfe method. They also investigate the $O(T^{-½})$ last-iterate convergence, which matches the best-known results in this context. Additionally, they present an $O(T^{-⅙})$ last-iterate convergence in the stochastic setting, which is, to their knowledge, the first last-iterate convergence result for solving constrained stochastic MVI problems.

**Strengths:**

* The paper is well-written and easy to follow.

* The authors establish the first last-iterate convergence result for solving constrained stochastic MVI problems.

* They demonstrate the last-iterate convergence result for the proposed generalized Frank-Wolfe method. Instead of using computer-aided proofs such as SOS and performance estimation, their analysis primarily relies on the Lyapunov argument.

**Weaknesses:**

See the Questions part.

**Questions:**

* From my understanding, the key idea of the Frank-Wolfe method lies in the linear minimization oracle (LMO), which has been shown to be more efficient in computation [1]. In the proposed generalized Frank-Wolfe method, the additional regularizer disrupts the linear structure of the minimization problem. If we choose the regularizer to be a quadratic function, then line 3 of your algorithm reduces to the projection onto the feasible set $\mathcal{X}$. Therefore, I would like to know if there are any advantages compared to those projection-based algorithms, such as EG/OGDA? Would it be possible to provide an explanation of it from both theoretical aspect and empirical performance?

* In line 808 of your proof, the second equality is from the update of $x^{t+1}$. I think it lost the $w^t$ part. I am not sure whether this ignored part will influence the convergence rate you derived.

* Could you please elaborate more on the introduction of the momentum step for gradient estimation $y^t$ in Algorithm 3? Is it essential for the convergence in the stochastic version? If so, could you provide the motivation for how it guarantees convergence?

[1] Combettes, Cyrille W., and Sebastian Pokutta. "Complexity of linear minimization and projection on some sets." Operations Research Letters 49.4 (2021): 565-571.

**Limitations:**

The convergence analysis heavily depends on the regularizer to ensure the Lipschitz property of the solutions of the minimization problem (see Lemma 4.1). When $\tau=0$, it reduces to the FW algorithms, and its analysis is not provided. The authors also mention this in the conclusion.

---

> ### Author Rebuttal · Authors · 2024-08-06
>
> >Comment: From my understanding, the key idea of the Frank-Wolfe method lies in the linear minimization oracle (LMO) ...
>
> **Response:** We agree with the reviewer that by adding a regularizer, one can no longer use an LMO to find the Frank-Wolfe direction. However, also due to the regularizer, the problem of finding the Frank-Wolfe direction becomes a strongly convex optimization problem, which can be solved efficiently, or even admits closed-form solutions (such as in zero-sum games). Therefore, the additional computational challenge in obtaining the Frank-Wolfe direction is not an issue.
>
> Compared with related projection gradient-based algorithms, such as EG, in the deterministic setting, we do not see any clear advantage, as they both achieve the same rate of convergence and similar empirical performance (see Appendix E Figure 1). In the stochastic setting, our method has a clear advantage since it is provably convergent while EG is shown to be unstable in general (see reference [15] in our work). In the next version of this work, we will provide a more extensive numerical study to compare the performance of generalized Frank-Wolfe to EG and OG.
>
> That being said, the goal of this paper was not to propose an algorithm that outperforms existing benchmarks. In recent years, existing studies of monotone variational inequalities have primarily focused on gradient-based algorithms such as EG and OG. On the other hand, Frank-Wolfe is also a natural algorithm (but is, to some extent, overlooked), and has a nice connection with fictitious play in zero-sum games. We study this algorithm and show that with a simple modification (that is, adding a regularizer), the convergence rate of Frank-Wolfe matches with that of EG and OG in the deterministic setting and is provably convergent in the stochastic setting.
>
> >Comment: In line 808 of your proof, the second equality is from the update of $x_{t+1}$. I think it lost the $w_t$ part. I am not sure whether this ignored part will influence the convergence rate you derived.
>
> **Response:** We greatly appreciate the reviewer for carefully reading our work. The missing $w_t$ in Line 808 was indeed a typo. Next, we present a fix to this issue. We start with the decomposition of the term $E_1$ in Line 808:
>
> $$E_1=(s_t-s_{t+1})^\top (F(x_{t+1})-F(x_t))+(x_t-s_t)^\top (F(x_{t+1})-F(x_t))\quad\qquad (1)$$
>
> For the first term on the right-hand side of the previous inequality, we have by Assumption 4.1 and Lemma 4.1 that
> $$
> (s_t-s_{t+1})^\top (F(x_{t+1})-F(x_t))\leq \vert|s_t-s_{t+1}\vert|\vert|F(x_{t+1})-F(x_t)\vert|
> \leq \frac{L^2}{\tau \sigma_f}\vert|x_{t+1}-x_t\vert|^2.\quad (2)
> $$
> This is the same as what we did in Line 808 (from the first inequality to the third inequality).
>
> The typo was with our approach in bounding the second term on the right-hand side of Eq. (1). Next, we present a different approach to bound it, which requires the following additional but mild assumption.
>
> **Assumption:** The Jacobian matrix $J(\cdot)$ of the operator $F(\cdot)$ is Lipschitz continuous.
>
> Since $F(\cdot)$ is Lipschitz continuous (Assumption 4.1), its Jacobian matrix is well-defined almost everywhere. Assumption imposes a mild smoothness condition on the Jacobian matrix, which is automatically satisfied in the zero-sum game setting, where the operator $F(\cdot)$ is a linear operator. Note that the monotonicity of $F(x)$ implies that $J(x)+J(x)^\top$ is a positively semidefinite matrix for any $x\in\mathcal{X}$.
>
>
> Denote the Lipschitz constant of $J(\cdot)$ as $L_J$. To proceed, in view of the second term on the right-hand side of Eq. (1), for a given $v\in\mathbb{R}^d$, consider the function $g_v(x)=v^\top F(x)$. Since
> $$\vert|\nabla g_v(x_1)-\nabla g_v(x_2)\vert|=\vert|(J(x_1)-J(x_1))^\top v\vert|\leq L_J\vert|v\vert|\vert|x_1-x_2\vert|,$$
> the function $g_v(x)$ is an $L_J\vert|v\vert|$-smooth function with respect to $x$. Therefore, using the descent lemma, we have for any $x_1,x_2\in\mathcal{X}$:
> $$g_v(x_2)-g_v(x_1)\leq (x_2-x_1)^\top J(x_1)^\top v+\frac{L_J\vert|v\vert|}{2}\vert|x_1-x_2\vert|^2.$$
> Identifying $v=x_t-s_t$, $x_1=x_t$, and $x_2=x_{t+1}$ in the above inequality, the second term on the right-hand side of Eq. (1) can be bounded as follows:
> $$(x_t-s_t)^\top (F(x_{t+1})-F(x_t))\qquad\qquad\qquad\qquad\qquad\qquad\qquad\qquad\qquad\qquad\qquad$$
> $$\leq (x_{t+1}-x_t)J(x_t)^\top (x_t-s_t)+\frac{L_J\vert|x_t-s_t\vert|\vert|x_{t+1}-x_t\vert|^2}{2}\qquad\qquad\qquad\qquad\qquad\qquad\qquad$$
> $$= -\alpha_t(x_t-s_t+w_t)J(x_t)^\top (x_t-s_t)+\frac{L_J\vert|x_t-s_t\vert|\vert|x_{t+1}-x_t\vert|^2}{2}\qquad\qquad\qquad\qquad\qquad$$
> $$\leq -\alpha_t(x_t-s_t)^\top \left[\frac{J(x_t)+J(x_t)^\top}{2}\right] (x_t-s_t)-\alpha_tw_t^\top J(x_t)^\top (x_t-s_t)+\frac{2L_JD_{\mathcal{X}}}{2}\vert|x_{t+1}-x_t\vert|^2$$
> $$\leq -\alpha_tw_t^\top J(x_t)^\top (x_t-s_t)+\frac{2L_JD_{\mathcal{X}}}{2}\vert|x_{t+1}-x_t\vert|^2\qquad\qquad\qquad\qquad\qquad\qquad\qquad\qquad(3)
> $$
> where the first equality follows from the update equation for $x_t$, the second inequality follows from $\vert|x_t-s_t\vert|\leq \vert|x_t\vert|+\vert|s_t\vert|\leq 2D_{\mathcal{X}}$ (recall that $D_{\mathcal{X}}$ is the radius of the feasible set), and the last inequality follows from $J(x_t)+J(x_t)^\top$ being a positively semidefinite matrix (due to the monotonicity of $F(\cdot)$).

---

> ### Author Response · Authors · 2024-08-06
> **Rebuttal (Continued)**
>
> Combining Eqs. (2) and (3) with Eq. (1), assuming without loss of generality that the regularization parameter $\tau$ is chosen to be less than $1$, we have
> $$
> E_1=(s_t-s_{t+1})^\top (F(x_{t+1})-F(x_t))+(x_t-s_t)^\top (F(x_{t+1})-F(x_t))$$
> $$\leq -\alpha_tw_t^\top J(x_t)^\top (x_t-s_t)+\left(\frac{2L_JD_{\mathcal{X}}}{2}+\frac{L^2}{\tau \sigma_f}\right)\vert|x_{t+1}-x_t\vert|^2$$
> $$\leq -\alpha_tw_t^\top J(x_t)^\top (x_t-s_t)+\left(\frac{2L_JD_{\mathcal{X}}}{2}+\frac{L^2}{\sigma_f}\right)\frac{1}{\tau}\vert|x_{t+1}-x_t\vert|^2$$
> $$= -\alpha_tw_t^\top J(x_t)^\top (x_t-s_t)+ \frac{M}{\tau}\vert|x_{t+1}-x_t\vert|^2,\qquad\qquad\qquad(4)
> $$
> where we denote $M=\frac{2L_JD_{\mathcal{X}}}{2}+\frac{L^2}{\sigma_f}$ in Eq. (4) for simplcity of notation. To proceed, using the same line of analysis as we did from the third inequality to the last inequality in Line 808, we have
> $$
> E_1
> \leq -\alpha_tw_t^\top J(x_t)^\top (x_t-s_t)+\frac{M}{\tau}\vert|x_{t+1}-x_t\vert|^2\qquad\qquad\qquad$$
> $$\leq -\alpha_tw_t^\top J(x_t)^\top (x_t-s_t)+\frac{8\alpha_t^2MD_{\mathcal{X}}^2}{\tau} +\frac{2M\alpha_t^2}{\tau}\vert|w_t\vert|^2.
> $$
> In view of the previous inequality, the first term on the right-hand side is mean zero due to $\{w_t\}$ being a martingale difference sequence (cf. Assumption 4.2), and the rest of the terms are the same (up to a multiplicative constant that is independent of $\tau$) as the terms on the right-hand side of the last inequality in Line 808. Following the same analysis starting from Line 809, the overall finite-time bound (with the same rate of convergence) can be reproduced.
>
> **In summary**, the convergence analysis for Algorithm 2 can be reproduced with the additional assumption that the Jacobian matrix of $F(\cdot)$ is Lipschitz continuous. The analysis of Algorithm 1 for zero-sum games remains intact since the operator $F(\cdot)$ in that case is linear, hence the Jacobian is a constant, which is automatically Lipschitz. The analysis for Algorithm 3 remains intact as it does not require the additional assumption.
>
> It is an interesting future direction to investigate whether the additional Lipschitz continuity assumption on the Jacobian matrix is necessary for the convergence of Algorithm 2. Intuitively, when there is stochasticity in the algorithm, we need some form of smoothness in the update to control the stochastic error. For Algorithm 3, this is achieved through the algorithm design (by creating a timescale separation between the variance reduced stochastic estimate $y_t$ and the main iterate $x_t$). In Algorithm 2, since the noise is directly added to the update equation, we imposed the additional Lipschitz continuity assumption for the analysis.

---

> > ### Comment · Reviewer_UPhr · 2024-08-10
> >
> > Thank you for the response. I appreciate the rebuttal, as it has effectively addressed some concerns, particularly regarding the gap in the proof. I will adjust the score accordingly. However, I believe more effort is needed to remove the new introduced assumption.

---

> > > ### Author Response · Authors · 2024-08-12
> > > **Follow-Up Comments by Authors**
> > >
> > > We are glad to hear that our response addresses some of the reviewer’s concerns, especially regarding the proof gap, and we greatly appreciate the reviewer for raising the score. Next, we would like to further justify our newly imposed assumption, which does not undermine the contributions of this work. First, it does not imply any strong curvature properties (such as strong monotonicity of the operator or strong convexity of the feasible set). Moreover, our original motivation for allowing the update to be noisy in Algorithm 2 is that it can directly be used to obtain the convergence bound of smoothed fictitious play, where the newly imposed assumption is automatically satisfied due to the linear structure of the operator $F(\cdot)$. In either the completely deterministic setting (where $w_t=0$) or the more challenging stochastic setting (where $F(x_t)$ is noisy), both our original approaches work.

---

> ### Author Response · Authors · 2024-08-06
> **Rebuttal (Continued)**
>
> >Comment: Could you please elaborate more on the introduction of the momentum step for gradient estimation $y_t$ in Algorithm 3? Is it essential for the convergence in the stochastic version? If so, could you provide the motivation for how it guarantees convergence?
>
> **Response:**  Algorithm 3 Line 3 is indeed a key step in the algorithm design and is essential for the convergence analysis. In the following, we first consider the case without the variance-reduced gradient estimation step, illustrate the potential issues, and then elaborate on why introducing Algorithm 3 Line 3 addresses this issue.
>
> Suppose that we directly use the noisy estimate $F(x_t)+z_t$ in Algorithm 3 Line 4 to compute the Frank-Wolfe direction (instead of going through Algorithm 3 Line 3 to construct the variance-reduced estimate $y_t$). Note that, despite replacing the hardmin with a softmin (by introducing the regularizer), the algorithm is extremely sensitive to the noise $z_t$ because the Frank-Wolfe direction computed from the accurate $F(x_t)$ and the noisy version $F(x_t)+z_t$ could be completely different. As a result, due to the lack of control on the noise, there is no convergence of $\{x_t\}$.
>
> Now, consider Algorithm 3 Line 3. Recall that we created a timescale separation between $y_t$ and $x_t$ by choosing $\beta\gg \alpha$. Therefore, from the perspective of $y_t$, $x_t$ is close to being stationary. For the purpose of illustration and for understanding the update equation of $y_t$, suppose that $x_t$ is completely stationary (which corresponds to $\alpha=0$). In this case, we will drop the subscript $t$ and just write $x_t$ as $x$. Given a sequence of noisy estimates ${(F(x)+z_i)}$, $0\leq i\leq t$, of $F(x)$, by running Algorithm 3 Line 3, $y_t$ is essentially a convex combination (hence a weighted average) of all the historical noisy estimates $( F(x)+z_i )_{0\leq i\leq t}$. Therefore, $y_t$ is an unbiased estimate of $F(x)$ with a much smaller variance compared with $F(x)+z_t$. As a result, compared with directly using $F(x)+z_t$, the Frank-Wolfe direction computed from using $y_t$ in Algorithm 3 Line 4 is a much more accurate estimate of the Frank-Wolfe direction computed from using the accurate $F(x)$, which eventually leads to the convergence of Algorithm $3$.
>
> As a final remark, we assumed $x_t$ being stationary from the perspective of $y_t$ in the above illustration, while both $x_t$ and $y_t$ are moving simultaneously (while at different rates) in Algorithm 3. Therefore, the proof consists of a rigorous two-timescale analysis by studying both the estimation error $\vert|y_t-F(x_t)\vert|^2$ and the gap of the Lyapunov function $V(\cdot)$ at the same time.
>
> >Comment: The convergence analysis heavily depends on the regularizer to ensure the Lipschitz property of the solutions of the minimization problem (see Lemma 4.1). When $\tau=0$, it reduces to the FW algorithms, and its analysis is not provided. The authors also mention this in the conclusion.
>
> **Response:** The introduction of the regularizer is indeed crucial for the analysis. When $\tau=0$, even in the case of fictitious play for zero-sum games (which is a special case of our setup), establishing the $O(1/\sqrt{T})$ rate of convergence has been an open problem for more than 60 years. This is called the Karlin’s weak conjecture [1].  Resolving this open problem (and extending it to Frank-Wolfe for solving general MVI problems) is the ultimate goal of this line of research.
>
> >[1] Karlin, S. (2003). Mathematical methods and theory in games, programming, and economics (Vol. 2). Courier Corporation.

---

> ### Author Response · Authors · 2024-08-09
> **Rebuttal Feedback**
>
> We greatly appreciate the reviewer’s time and effort in evaluating our work. Please let us know if our response addresses the concerns raised, or if further clarification is needed.

---

### Official Review · Reviewer_EbDg · 2024-07-10

**Soundness:** 4
**Presentation:** 4
**Contribution:** 4
**Rating:** 9
**Confidence:** 4

**Summary:**

This paper presents combines Frank-Wolfe algorithm and entropy regularization trick to propose a generalized Frank-Wolfe algorithm for solving MVI problems. The paper proves $O(T^{-1/2})$ last-iterate convergence rate for the deterministic case. And it extends the algorithm to stochastic MVIs with $O(T^{-1/6})$ last-iterate convergence rate.

**Strengths:**

1. I think it's a good idea to combine entropy regularization with FW algorithm.

2. The paper is very well written and easy to follow. The intuition for the proposed algorithms is well explained. The assumptions are standard and clearly listed. The dependence of the convergence rates on everything is explicitly written out.

3. Strong theoretical guarantees are provided in this paper. For example, it gives the first last-iterate convergence guarantee for constrained stochastic MVIs without strong curvature assumptions. It also gives the first finite-time analysis of smoothed FP.

4. Numerical experiments are conducted to demonstrate the efficiency of the proposed algorithms.

**Weaknesses:**

I think there's no apparent weaknesses. I suspect the analysis for the stochastic case might not be tight, leaving room for improvement.

**Questions:**

-

**Limitations:**

Yes.

---

> ### Author Rebuttal · Authors · 2024-08-05
>
> >Comment: I think there's no apparent weaknesses. I suspect the analysis for the stochastic case might not be tight, leaving room for improvement.
>
> **Response:** We greatly appreciate the reviewer acknowledging the contributions of our work. We agree with the reviewer that the convergence rate of $O(1/T^{1/6})$ in the stochastic setting is likely sub-optimal. Improving the convergence rate (by algorithm design or by analysis) is an immediate future direction of this work.

---

### Official Review · Reviewer_BD4Y · 2024-07-11

**Soundness:** 4
**Presentation:** 4
**Contribution:** 3
**Rating:** 6
**Confidence:** 4

**Summary:**

This paper proposes a Frank-Wolfe algorithms for solving monotone VI problems, for both deterministic and stochastic settings, proof that the first one matches the optimal convergence rate, and provide last-iterate convergence guarantees for the second one, which is a novelty if no curvature requirement is imposed on neither set nor operator.

**Strengths:**

The analogy between classical game theory algorithm and Frank-Wolfe is simple yet creative and fruitful for providing theoretical guarantees for the former one. The operator smoothing approach in the design of the stochastic algorithm is also elegant and effective. I consider this paper worth reading thanks to these theoretical findings. Numerical experiments are also needed, but they show no significant advantage over alternative approaches, so the contribution of the paper seems only theoretical.

**Weaknesses:**

Despite the theoretical focus of the paper, it is recommended to make experiments more contentful by considering other problems, at least games like Burglar-Policeman matrix game, or some more advanced example of VI, maybe involving the adversarial NN. Regarding the theory, I see no significant flaws.

**Questions:**

Typos and notation related remarks:
- page 1, line 19: \to is more convenient than \mapsto
- page 2, sentence at line 91: "the algorithm has been shown to diverge" is unclear, you have just mentioned the algorithm you have proposed, and it does not diverge, so which one does?
- page 3, line 106: in phrase "in either the uncontrained regime [13 ] or under", "in" should be after "either", and "uncontrained" has typo
- page 4, line 146: subscript and superscript for A are confused
- Condition 4.1 allows one to remove limit s \in X of max in line 3 of Algorithm 2, at least in accordance with line 217
- page 6, line 228: "make ... assumptions" or "impose ... requirements"

Besides, the "shadow" indicating the std of function's values from multiple random runs should be added to the plots with stochastic Frank-Wolfe method.

**Limitations:**

The limitations are clear from reading.

---

> ### Author Rebuttal · Authors · 2024-08-05
>
> >Comment: Despite the theoretical focus of the paper, it is recommended to make experiments more contentful by considering other problems ...
>
> **Response:** We thank the reviewer for the suggestion. In the next version, we will include an additional numerical example to demonstrate the performance of the generalized Frank-Wolfe algorithm relative to other algorithms.
>
> >Comment: Typos and notation related remarks.
>
> **Response:** We thank the reviewer for carefully reading our work. We will fix the typos and notation-related problems on pages 1, 3, 4, Condition 4.1, and page 6 as pointed out by the reviewer. Regarding the sentence ‘the algorithm has been shown to diverge’ in line 91 of page 2, it refers to the extragradient algorithm. We will clarify this in the next version of our work.
>
>
> >Comment: Besides, the "shadow" indicating the std of function's values from multiple random runs should be added to the plots with stochastic Frank-Wolfe method.
>
> **Response:** We will plot the standard deviation for the stochastic Frank-Wolfe algorithm in our numerical simulations.

---

> ### Author Response · Authors · 2024-08-09
> **Rebuttal Feedback**
>
> We greatly appreciate the reviewer’s time and effort in evaluating our work. Please let us know if our response addresses the concerns raised, or if further clarification is needed.

---

### Author Response · Authors · 2024-08-05
**Author's Response**

We thank all reviewers for carefully reading our work. In the following, we provide a point-by-point response to all reviewers' comments.

---

### Decision · Program_Chairs · 2024-09-25

**Decision:**

Accept (poster)

**Comment:**

The reviewers are unanimous in favor of this submission.